# NeuSTIP: A Neuro-Symbolic Model for Link and Time Prediction in Temporal Knowledge Graphs

**Ishaan Singh    Navdeep Kaur    Garima Gaur    Mausam**
Indian Institute of Technology, Delhi
{ishaanyuvraj, navdeepkjohal, garimagaur27}@gmail.com
mausam@cse.iitd.ac.in

## Abstract

Neuro-symbolic (NS) models for knowledge graph completion (KGC) combine the benefits of symbolic models (interpretable inference) with those of distributed representations (parameter sharing, high accuracy). While several NS models exist for KGs with static facts, there is limited work on *temporal* KGC (TKGC) for KGs where a fact is associated with a time interval. In response, we propose a novel NS model for TKGC called NeuSTIP, which performs link prediction and time interval prediction in a TKG. NeuSTIP learns temporal rules with Allen predicates, which ensure temporal consistency between neighboring predicates in the rule body. We further design a unique scoring function that evaluates the confidence of the candidate answers while performing link and time interval predictions by utilizing the learned rules. Our empirical evaluation on two time interval based TKGC datasets shows that our model shows competitive performance on link prediction and establishes a new state of the art on time prediction.

## 1  Introduction

Knowledge Graphs (KGs) are factual information repositories, where each *fact* is encoded as $r(s, o)$, where $s$ and $o$ are the real-world entities and $r$ is the relationship between them. For instance, the fact *presidentOf(Joe Biden, USA)* represents the fact that *Joe Biden* is the president of *USA*. Temporal KGs extend these to entity-entity relations that have a temporal facet, for e.g., *workedAt(Einstein, ETH_Zurich, [1912,1914])*. In this work, we study Temporal KGs that maintain temporal facts, $r(s, o, T)$, annotating each fact $r(s, o)$ with the time period $T$ for which it holds.

While being a popular source of structured information, KGs are often incomplete. To this end, the problem of enriching KGs by inferring missing information is formulated as a KG completion (KGC) task. In the context of static (non-temporal)

KGs, a key KGC task is *link prediction*, viz, given a query $r(s, ?)$, predict $o$ for which the fact $r(s, o)$ holds in the real-world. This problem is fairly well-studied and has been tackled in many different ways. Existing works can be broadly categorized into GNN-based solutions (Zhu et al., 2021; Vashishth et al., 2020), LM-based approaches (Yao et al., 2019; Wang et al., 2022), KG embedding-based models (Sun et al., 2019; Trouillon et al., 2016), and neuro-symbolic (NS) solutions (Yang et al., 2017; Qu et al., 2021). Of special interest to us are NS approaches, which bring together human-interpretable deduction capabilities of symbolic models with parameter sharing and other benefits of distributed representations in neural models.

Temporal KG completion (TKGC) extends KGC task to TKGs. In addition to link prediction, it involves an additional task of *time-interval* prediction – given a query $r(s, o, ?)$, infer the time interval when the fact holds true. The majority of existing solutions are KG-embedding based (Dasgupta et al., 2018; Jain et al., 2020; Xu et al., 2020), and do not incorporate symbolic rules. NS approaches have received very limited attention for TKGC. TLogic and ALRE-IR (Liu et al., 2021; Mei et al., 2022) study NS-TKGC, but for time-instant KGs (i.e., KGs where a fact associated with an instant, not an interval). TILP (Xiong et al., 2023) is a very recent model for time-interval TKGC, but it can only do link prediction. To the best of our knowledge, no NS-TKGC approach exists that performs both link and time-interval prediction.

A key challenge in building an NS-TKGC model is to design a unified rule language, which is intuitive and interpretable to humans, and also effective on both prediction tasks. Secondly, ideally, the confidence of a rule should be computed based on both the statistical properties of the rule groundings and also the similarity scores of latent representations (which is not the case in existing models).

**Contributions:**  We  propose  NeuSTIP (**Neu**ro

Symbolic Link and Time Interval Prediction), the first comprehensive NS framework for TKGC, which is effective on both link prediction and time interval prediction tasks. It uses an intuitive rule language that integrates the complete set of Allen algebra relations and KG relations, with the goal of enforcing temporal consistency between neighboring predicates in the rule body. It also uses a novel way to compute confidence of temporal rules that combines both symbolic and embedding information. Moreover, to the best of our knowledge, it is the first NS-TKGC model to perform time-interval prediction.We evaluate the performance of NeuSTIP using two benchmark time interval KGC datasets, WIKIDATA12k and YAGO11k (Dasgupta et al., 2018). We find that a hybrid of NeuSTIP with a KG-embedding model TimePlex (Jain et al., 2020) obtains best results on both prediction tasks in both datasets. We release the NeuSTIP[1] code-base for further research.

## 2 Related Work

There are three main types of TKGC models in literature: purely embedding-based, multi-hop reasoning based, and rule-based.

**Embedding-based Models:** All these models learn embeddings of entities and relations and define an associated scoring function to assess the validity of a temporal fact. Earlier methods incorporated time within entity embeddings (Dasgupta et al., 2018; García-Durán et al., 2018), and used TransE-based scoring (Bordes et al., 2013). Recent methods (Messner et al., 2022; Sadeghian et al., 2021) adapt static KGE models such as RotatE (Sun et al., 2019), BoxE (Abboud et al., 2020) for TKGC task. Other approaches such as TransE-TAE (Jiang et al., 2016), Timeplex (Jain et al., 2020) explicitly learn time embeddings, and additionally model temporal constraints between pairs of tuples in the TKG. These models are generally less interpretable, due to their complete reliance on latent embeddings.

**Multi-hop Reasoning Models:** Such models exploit neighborhood information of an entity by employing distinct graph neural network architectures (Kipf and Welling, 2017). Models such as TeMP (Wu et al., 2020), RE-NET (Jin et al., 2020), xERTE (Han et al., 2021a), CyGNet (Zhu et al., 2020) exploit self-attention/GRU, RNN, time-aware attention mechanism, and Copy-Generation

---

[1] https://github.com/dair-iitd/NeuSTIP.git

model respectively to integrate time information in a GNN. GNN-based models are generally computationally expensive (Luo et al., 2022), and do not scale well to large datasets.

**Neuro-Symbolic (Rule-based) Models:** These models combine neural embeddings with explicit rule learning and inference, getting benefits of both interpretable inference of symbolic models and parameter sharing of neural models. TLogic (Liu et al., 2021) and ALRE-IR (Mei et al., 2022) are NS link-prediction models designed for time-instant TKGs. Probably the closest to our work is a very recent model TILP (Xiong et al., 2023) that performs all possible constrained walks on time interval datasets to learn temporal logic rules and adopts attention mechanism to score each rule. There are two key differences between TILP and NeuSTIP. Firstly, TILP's rules require expressing temporal relations between all pairs of intervals mentioned in the rule body. This blows up the number of rules, forcing TILP to use only three Allen relations. In contrast, NeuSTIP encodes temporal relations only between pairs of neighboring intervals in a rule path, and uses all 13 Allen relations. Our experiments show that NeuSTIP performs better inferences compared to TILP. Secondly, and more importantly, TILP is developed only for link prediction, whereas NeuSTIP introduces specific score and loss functions for time-interval prediction.

### 2.1 Time-Interval Prediction

Time prediction in TKG is relatively underexplored. Existing research includes Know-Evolve (Trivedi et al., 2017) and GHNN (Han et al., 2020), which perform *time instant* prediction by modeling a given TKG fact as a point process. Time-interval prediction is studied in embedding-based models such as Time2Box (Cai et al., 2021) and Timeplex (Jain et al., 2020) models – they develop novel TKGE-based scoring functions for the task. To the best of our understanding, no NS TKGC method exists for time-interval prediction, a gap that NeuSTIP aims to fill.

## 3 Preliminaries

A knowledge graph maintains a set of entities $\mathcal{E}$, and relations $\mathcal{R}$, and each fact $r(s, o)$ is denoted by a directed edge from subject entity $s$ to object entity $o$, with the label $r$. A *temporal* KG (TKG) $\mathcal{K}$ additionally maintains the time domain, which is suitably discretized into a set $\mathcal{T}$ of discrete time

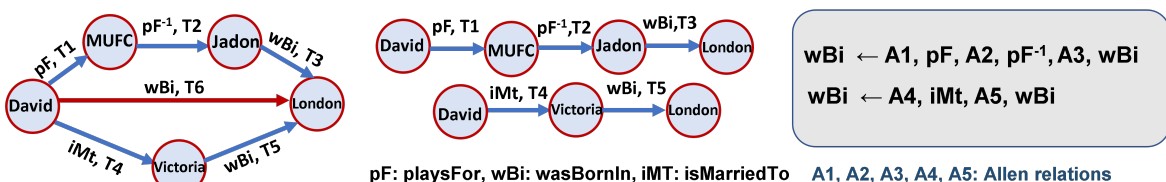

Figure 1: Extracting rules using the target fact $wBi(David, London, T6)$: Traversing KG (left) to find all paths (center) from *subject* entity $David$ to *object* entity $London$; Extracting one rule per grounding (center) interleaving Allen relations $A_\bullet$ that specify relations between time intervals of neighboring KG relations in rule body.

instants, with two special elements $t_{min}$ and $t_{max}$ denoting the minimum and maximum time instants attainable in the TKG. A temporal fact is $r(s, o, T)$, where $T = [t_b, t_e]$ denotes the time interval during which the relation was true ($t_b, t_e \in \mathcal{T}$). It is represented in the TKG as an edge $s \xrightarrow{(r, T)} o$ between entities $s$ and $o$ (see Figure 1 (left)). For ease of model design, a common pre-processing step increments the TKG by adding inverse relations (and therefore inverse edges) to the graph (Jain et al., 2020). Any TKG Completion model is evaluated using (1) link prediction queries: tail prediction, $r(s, ?, T)$ and head prediction, $r^{-1}(o, ?, T)$, and (2) time interval prediction queries: $r(s, o, ?)$.

### 3.1 Background on Allen Algebra

Our model is based on Allen's interval calculus (Allen, 1983), which is a formal system to represent relations between time intervals. It defines 13 exhaustive and pairwise-disjoint relation types. E.g., given two time intervals $T1 = [t1_b, t1_e]$ and $T2 = [t2_b, t2_e]$, the Allen relation `overlaps(T1, T2)` holds if $t1_b < t2_b < t1_e < t2_e$ (See Appendix A for more details). To avoid ambiguity, we refer to relations between time intervals as *Allen relations* and relations $r \in \mathcal{R}$ between entities as *KG relations*. Our model uses all 13 Allen relations in its rule language.

## 4 The Proposed NeuSTIP Model

We first describe NeuSTIP's rule language and its algorithm for mining rules. We then discuss its model for scoring a candidate answer, for both link and time interval prediction queries (See Figure 2). Further, we describe the loss function and inference procedures.

### 4.1 Mining of Temporal Rules

NeuSTIP learns first-order logic rules of the form:

$$r_h(e_1, e_{m+1}, T_1) \longleftarrow \wedge_{i=1}^{m} \big(a_i(T_i, T_{i+1}) \\ \wedge r_i(e_i, e_{i+1}, T_{i+1})\big) \quad (1)$$

where $a_i$ and $r_i$ denote the *Allen relations* and *KG relations* respectively, and $e_i$ and $T_i$ are *variables* that will ground in the entity set $\mathcal{E}$ and time interval set $\mathcal{T} \times \mathcal{T}$; $m$ is the rule length. The rule body can be seen as a path from $e_1$ to $e_{m+1}$, where each KG relation $r_i$ shares the *object* entity with the *subject* entity of relation $r_{i+1}$, $1 \le i < m$. Similarly, Allen relations $a_i$ specify relations between two consecutive time intervals, with the first Allen predicate $a_1(T_1, T_2)$ of the rule body being a relation between the time interval in the rule head and the first time interval mentioned in the body. Refer to Appendix B for examples of temporal rules.

NeuSTIP mines temporal rules by, for each known fact $r_h(s, o, T) \in \mathcal{K}$, finding ground paths in TKG from $s$ to $o$ of length up to a max limit. For instance, as shown in the center of Figure 1, for the fact $wBi(David, London, T6)$, all paths from *David* to *London* are found. Each ground path gets converted to exactly one first-order rule (by replacing entities and time intervals with variables), based on the specific KG relations in the path, and the specific Allen relations between each pair of time intervals (as shown in Figure 1). We highlight that our rule extraction process does not involve sampling walks, rather we use *all* walks – a departure from existing non-temporal rule mining approaches (Qu et al., 2021). We choose this because our preliminary analysis suggested that TKGs are sparser than non-temporal KGs, hence, sampling or random walks may miss important rules.

### 4.2 Scoring Candidate Answers

NeuSTIP first finds candidate answers $c$ and then scores each answer. For a link prediction query

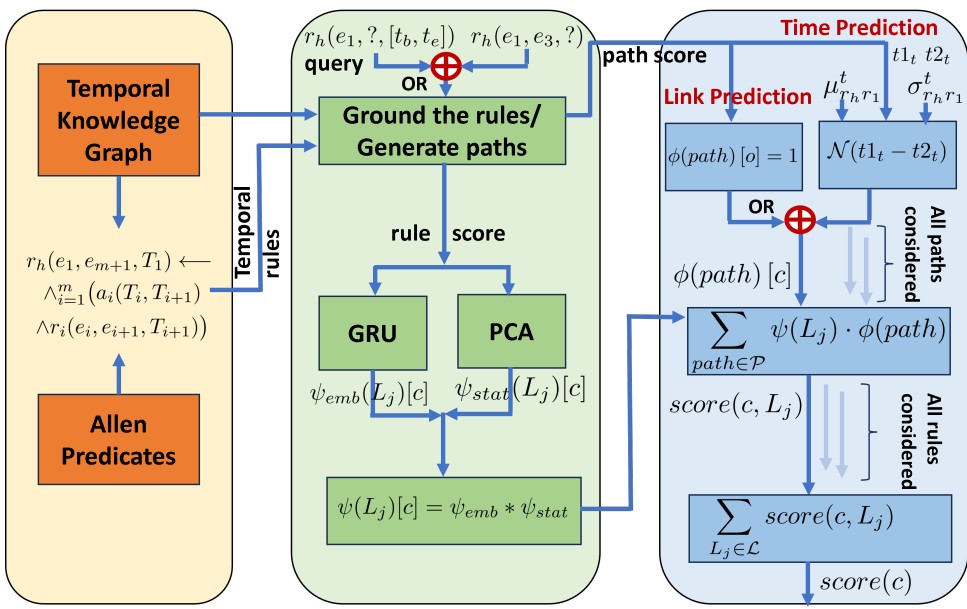

Figure 2: Overview of NeuSTIP evaluating link prediction $- r_h(e_1, ?, T_1)$ and time prediction $- r_h(e_1, e_3, ?)$ queries: Given a TKG and the generated rules (as shown in Figure 1), NeuSTIP first (middle) grounds the relevant rules and finds candidate answer $c$; later it computes rule score (middle) and corresponding path score (right) based on the type (link or time) of the query. The score rule computation involves evaluating embedding-based score $\psi_{emb}(L_j)[c]$ and statistical measure based $\psi_{stat}(L_j)[c]$. For scoring paths, link prediction queries treat all the paths similarly and assign a score of 1, whereas for time prediction NeuSTIP relies on priors (difference in start time and end time of pair of relations) learned from the data.

$r_h(s, ?, T)$, finding candidates is straightforward: find all relevant rules, identify each rule's groundings, and mark groundings of variable $e_{m+1}$ as candidate answer entities. However, for interval prediction $r_h(s, o, ?)$, it grounds all the relevant rules, by ignoring the first Allen relation $a_1(T_1, T_2)$. This is because it cannot be a priori ground, since $T_1$ is unknown (the intent is to predict it). It then identifies all time intervals $T_1$, which are consistent with the predicate $a_1(T_1, T_2)$, and outputs those as candidate answers. For example, if $a_1 = \texttt{before}$ and $T_2 = [1990, 2000]$, then it will output intervals $[t_b, t_e]$ such that $t_b < t_e < 1990$.

For interval prediction queries, NeuSTIP scores start and end time instants separately. Its scoring function is motivated by RNNLogic (Qu et al., 2021), and extended to temporal setting: the score of a candidate $c \in \{o, t_b, t_e\}$ is computed as

$$score(c) = \sum_{L_j \in \mathcal{L}} score(c, L_j) =$$
$$\sum_{L_j \in \mathcal{L}} \sum_{path \in \mathcal{P}} \psi(L_j)[c] \cdot \phi(path)[c] \quad (2)$$

Here, $\mathcal{L}$ is the set of all first-order rules $L_j$ that generate answer $c$, and $\mathcal{P}$ is the set of ground paths of

$L_j$ consistent with $c$. Note that a candidate answer $c$ can be arrived at by firing multiple rules in the rule set $\mathcal{L}$. Further, a given rule $L_j \in \mathcal{L}$ can be grounded to different paths which when followed in TKG arrive at candidate $c$. Our scoring function (in Equation 2) takes into account the significance of all such rules and the corresponding paths that lead to the generation of candidate $c$ by aggregating *rule score* $\psi(L_j)[\cdot]$ and *path score* $\phi(path)[\cdot]$. Since $c$ is of three types, the score of each rule $L_j$ has three components $\psi(L_j)[o]$, $\psi(L_j)[t_b]$ and $\psi(L_j)[t_e]$ used to score object entities, start time instants and end time instants, respectively. Similarly, the path scores are also specified using three components. We now explain NeuSTIP model that computes the *rule score* $\psi(L_j)[\cdot]$ and *path score* $\phi(path)[\cdot]$.

### 4.2.1 Rule Score

NeuSTIP uses a novel neuro-symbolic rule scoring function (Equation 4) – it combines a rule embedding based score $\psi_{emb}$ and a statistical measure-based score $\psi_{stat}$ to compute $\psi$.

**Embedding based rule score:** NeuSTIP learns three embeddings for a KG rule head relation $r$:

$\mathbf{r}^O$, $\mathbf{r}^B$, and $\mathbf{r}^E$, used for scoring for *object* entity, start time instants, and end time instant candidates, respectively. It also computes an embedding $\mathbf{L_j}$ for each rule $L_j$ (motivated by Mei et al. (2022)) by passing the sequence of Allen and KG relations in $L_j$'s rule body through a Gated Recurrent Unit (GRU) (Cho et al., 2014). For a candidate answer $c$, depending upon its *type* (denoted as *object*: $O$, *start time*: $B$, *end time*: $E$), the rule score of $L_j$ is computed as the cosine similarity between the rule embedding $\mathbf{L_j}$ and the relevant rule head embedding $\mathbf{r_h^\bullet}$ ($\in \{\mathbf{r}_h^O, \mathbf{r}_h^B, \mathbf{r}_h^E\}$),

$$\psi_{emb}(L_j)[c] = sim(\mathbf{L_j}, \mathbf{r_h^\bullet}) \quad (3)$$

Further details on GRU model are in Appendix C.

**Statistical measure based score:** NeuSTIP leverages PCA score for each rule as an additional measure of its accuracy. This symbolic rule confidence metric acts as a prior of the rules and helps in informed initialization of the *rule score*. PCA score estimates the fraction of entities predicted by the rule that are known to be true at training time (Galárraga et al., 2013). See Appendix D for more information. We denote it by $\psi_{stat}(L_j)[c]$, based on which type of entity being predicted by the rule.

NeuSTIP computes the aggregate *rule score* as:

$$\psi(L_j)[c] = \psi_{emb}(L_j)[c] * \psi_{stat}(L_j)[c] \quad (4)$$

### 4.2.2 Path Score

For link prediction, the model exploits a relatively simple approach and sets $\phi(path)[o] = 1$ for each $path$ that reaches the target answer $o$. This implies that the score of candidate entities is dependent on the total number of groundings and the quality of the corresponding rules. A low-quality rule with a less number of groundings cannot generate a high-scoring candidate entity.

For time prediction, a rule can only be partially ground. Notice the first Allen relation $a_1$ specifies the Allen relation between the time interval $T_1$ of $r_h$ and $T_2$ of $r_1$ (See Equation 1). Since $T_1$ is what we intend to predict, thus condition $a_1(T_1, T_2)$ cannot be ground. However, Allen relation $A_1$ and time interval $T2$ guide NeuSTIP (discussed in Section 4.2) to find its candidate start time $t1_b$ and end time $t1_e$ of time interval $T1$. For scoring the candidates, NeuSTIP computes two Gaussian distributions, $\mathcal{N}(\mu_{rr'}^B, \sigma_{rr'}^B)$ and $\mathcal{N}(\mu_{rr'}^E, \sigma_{rr'}^E)$ – they represent the start/end time difference of pairs of KG relations $(r, r')$ with the same subject entity.

Notice from Equation 1, relations $r_h$ and $r_1$ will capture two facts about the same subject $e_1$. Therefore, given the fact $r_1(e_1, e_2, [t2_b, t2_e])$, NeuSTIP calculates path score for start time candidate $t1_b$ in query $r_h(e_1, e_{m+1}, ?)$ as,

$$\phi(path)[t1_b] = \mathcal{N}(t1_b - t2_b | \mu_{r_h r_1}^B, \sigma_{r_h r_1}^B) \quad (5)$$

The same approach is used to estimate $\phi(path)[t1_e]$.

### 4.3 Loss Function

NeuSTIP is trained by minimizing two loss functions $\mathbf{L}_{LP}$ and $\mathbf{L}_{TP}$ for link prediction and interval prediction, respectively. The score of each candidate object entity $score(o)$ is normalized to $P(o)$ by computing softmax over the entity set $\mathcal{E}$. Similarly, the scores of start time and end time candidates are normalized over the time instant set $\mathcal{T}$. For a given query, we denote the set of known correct answers as $\mathbf{D}$ and the set of other answers as $\mathbf{N}$. We further use set $\mathbf{S}$ to denote the subset of $\mathbf{N}$ with a higher $score$ than that of the gold entity. For link prediction, the loss $\mathbf{L}_{LP}$ is computed as

$$\sum_{n \in \mathbf{N}} P(n) + \sum_{o \in \mathbf{D}} \left( \frac{\sum_{e \in \mathbf{S}}(P(e) - P(o))}{|\mathbf{S}|} \right) \quad (6)$$

Similarly, for time interval prediction let $\mathbf{D}$ denote the set of known true intervals. We construct two sets $\mathbf{D}_b$ and $\mathbf{D}_e$ with the start time instants and end time instants of intervals in $\mathbf{D}$. Further, $\mathbf{N}_b = \mathcal{T} \setminus \mathbf{D}_b$ and $\mathbf{N}_e = \mathcal{T} \setminus \mathbf{D}_e$. Because time instants $t_b$ and $t_e$ are numerical, NeuSTIP uses a loss function that captures differences between true and other times via $\mathbf{L}_{TP}$ as follows:

$$\sum_{t_b \in \mathbf{D}_b} \sum_{n_b \in \mathbf{N}_b} \left( P(n_b) - P(t_b) \right) * d(n_b, t_b) +$$
$$\sum_{t_e \in \mathbf{D}_e} \sum_{n_e \in \mathbf{N}_e} \left( P(n_e) - P(t_e) \right) * d(n_e, t_e) \quad (7)$$

where $d(\cdot, \cdot)$ is the time instant distance function. We define the distance function in Appendix E.

### 4.4 Inference

At the test time, we perform link prediction by ranking all objects $o$ based on $score(o)$. For time-interval prediction for query $r(s, o, ?)$, NeuSTIP constructs a $\mathcal{T} \times \mathcal{T}$ matrix, whose each entry is $P(t_b) * P(t_e) * \mathcal{N}(t_e - t_b | \mu_r^{intv}, \sigma_r^{intv})$ where $P(t_b)$ and $P(t_e)$ are the probabilities defined above, and

$\mathcal{N}(\cdot \,|\mu_r^{intv}, \sigma_r^{intv})$ is the Gaussian distribution over the duration of time-intervals of target relation $r$ in the query. After populating the matrix with data, we consider its upper-triangular matrix $\mathcal{TI}$ in order to ensure $t_e \geq t_b$. We return the predicted time interval as $[t_b, t_e] = \arg\max_{t_b,t_e}(\mathcal{TI})$.

## 4.5 Extended Model with Timeplex

Inspired by RNNLogic, we further ensemble our model with an embedding-based model, Time-Plex (Jain et al., 2020) to integrate the complementary features of NeuSTIP, our rule-based model and an embedding-based model as follows:

$$ens\_score(c) = score(c) + \eta * Timplex(c) \quad (8)$$

where $c \in \{o, t_b, t_e\}$ and $\eta$ is a learnable parameter. $score(c)$ is NeuSTIP score (Equation 2).

## 5 Experiments

### 5.1 Experimental Setup

**Datasets and Metrics:** We evaluate the proposed model on two standard time-interval TKGC datasets: WIKIDATA12k and YAGO11k (Dasgupta et al., 2018). For both datasets, we consider the temporal granularity to be 1 year. The dataset statistics are reported in Appendix F. For link prediction, we report the standard metrics of Mean Reciprocal Rank (MRR), Hits@1, and Hits@10 and use time-aware filtered measures in each model (Jain et al., 2020). For time interval prediction, we employ the aeIOU metric (Jain et al., 2020) as:

$$aeIOU(T^{ev}, T^{pr}) = \frac{max\{1, vol(T^{ev} \cap T^{pr})\}}{vol(T^{ev} \uplus T^{pr})}$$

such that $T^{ev}$ and $T^{pr}$ are gold and predicted time intervals, vol() refers to the size of the time interval (which would be in terms of number of years for these datasets), $T^{ev} \cap T^{pr}$ refers to overlap in time-interval, $T^{ev} \uplus T^{pr}$ is the smallest single contiguous interval containing $T^{ev}$ and $T^{pr}$.

**Baselines:** Link prediction in TKGC is relatively well-studied. We compare against 10 embedding and rule-based/neuro-symbolic solutions. Specifically, we choose a total of 4 rule based models – Neural-LP (Yang et al., 2017) and AnyBurl (Meilicke et al., 2019) handling static (non-temporal) KG, and TLogic (Liu et al., 2021) (time-instants) and TILP (Xiong et al., 2023) (time-intervals) working with temporal KGs. In embedding based approaches, we consider ComplEx (Trouillon et al.,

Table 1: Performance of models for link prediction task on WIKIDATA12k and YAGO11k.

| Model | WIKIDATA12k | | | YAGO11k | | |
|---|---|---|---|---|---|---|
| | MRR | H@1 | H@10 | MRR | H@1 | H@10 |
| Neural-LP | 18.23 | 9.08 | 38.48 | 10.01 | 4.01 | 18.45 |
| AnyBURL | 19.08 | 10.30 | 39.04 | 9.08 | 3.78 | 18.14 |
| TLogic | 25.36 | 17.54 | 44.24 | 15.45 | 11.80 | 23.09 |
| TILP-base | 31.14 | 21.52 | 50.77 | 18.80 | 13.36 | 30.89 |
| TILP | 33.28 | 23.42 | 52.89 | 24.11 | 16.67 | **41.49** |
| ComplEx | 24.82 | 14.30 | 48.90 | 18.14 | 11.46 | 31.11 |
| TA-ComplEx | 22.78 | 12.69 | 46.00 | 15.24 | 9.36 | 26.26 |
| HyTE | 25.28 | 14.70 | 48.26 | 13.55 | 3.32 | 29.81 |
| DE-SimplE | 25.29 | 14.68 | 49.05 | 15.12 | 8.75 | 26.74 |
| TNT-Complex | 30.10 | 19.73 | 50.69 | 18.01 | 11.02 | 31.28 |
| TimePlex (Base) | 32.38 | 22.03 | 52.79 | 18.35 | 10.99 | 31.86 |
| TimePlex | 33.35 | 22.78 | 53.20 | 23.64 | 16.92 | 36.71 |
| NeuSTIP (Base) | 31.98 | 22.11 | 50.31 | 23.81 | 17.26 | 35.15 |
| NeuSTIP w/ KGE | **34.78** | **24.38** | **53.75** | **25.23** | **18.45** | 37.76 |

2016), TA-ComplEx (García-Durán et al., 2018), HyTE (Dasgupta et al., 2018), DE-SimplE (Goel et al., 2020), TNT-Complex (Lacroix et al., 2020), and TimePlex (Jain et al., 2020). Apart from ComplEx (Trouillon et al., 2016), all the other embedding-based solutions are proposed especially for temporal KGs. The original TimePlex model itself has two versions – a base model and the full model. The full version adds two temporal consistency gadgets, modeling relation recurrence, and typical duration distributions between two relations. In the same vein, TILP also has two variants, and its full model additionally introduces temporal features such as recurrence, and duration distribution into the model. We compare against both versions of the TimePlex and TILP models.

For interval prediction, NeuSTIP is the first neuro-symbolic model – so there is no existing NS model to directly compare against. We resort to using only embedding-based models for comparisons: HyTE, TNT-Complex, and TimePlex.

We report results from two variants of our proposed model, NeuSTIP (base) and NeuSTIP with KGE. The base model is trained exclusively with the proposed temporal rules and corresponding candidate score, as in Equation 2. In NeuSTIP with KGE variant, we integrate the state-of-the-art KG embedding model, Timeplex (full) using an ensemble (see Section 4.5). The specifics of the training procedure of NeuSTIP and hyperparameter settings are reported in Appendix G, H and I. For all comparisons, where possible we report published results since our datasets are standard and exact splits have been used as is in earlier works.

Table 2: Performance of models for time interval prediction task on WIKIDATA12k and YAGO11k.

| Dataset | WIKIDATA12k | YAGO11k |
|---|---|---|
| Model | aeIOU | aeIOU |
| HyTE | 5.41 | 5.41 |
| TNT-Complex | 23.35 | 8.40 |
| Timeplex (Base) | 26.20 | 14.21 |
| Timeplex | 26.36 | 20.03 |
| NeuSTIP (Base) | 27.36 | 22.49 |
| NeuSTIP w/ KGE | **28.21** | **27.83** |

Table 3: Link prediction performance of NeuSTIP and variants, NeuSTIP-TR (w/o temporal constraints), NeuSTIP-PCA (w/o PCA score)

| Model | WIKIDATA12k | | | YAGO11k | | |
|---|---|---|---|---|---|---|
| | MRR | H@1 | H@10 | MRR | H@1 | H@10 |
| NeuSTIP | **31.98** | **22.11** | **50.31** | **23.81** | **17.26** | **35.15** |
| NeuSTIP- TR | 26.00 | 15.60 | 47.14 | 18.90 | 11.68 | 30.40 |
| NeuSTIP- PCA | 29.69 | 20.10 | 47.77 | 22.14 | 16.07 | 32.72 |

## 5.2 Results and Observations

**Link Prediction:** We report the performance of NeuSTIP and other baseline methods in Table 1. We observe that the performance of our NeuSTIP (base) is comparable with the base model of Time-Plex for WIKIDATA12k, while the gap is more pronounced on YAGO11k as even our base model outperforms TimePlex (base) on all the three metrics with a gain of over 4 MRR pts. Similarly, our base model is comparable to the base model of the best temporal rule-based model (TILP-base) for WIKIDATA12k whereas our base model outperforms it on all three metrics on the YAGO11k dataset. We also observe that our 'NeuSTIP w/ KGE' model, outperforms the state-of-the-art models on 2 out of 3 metrics on YAGO11k, and on all metrics on WIKIDATA12k.

**Time Interval Prediction** Table 2 reports the results of time interval prediction. We observe that the performance of our base model (NeuSTIP (Base)) is better than that of all models on both datasets. Our 'NeuSTIP w/ KGE' model yields further improvements, outperforming the state-of-the-art Timeplex model on YAGO11k by a strong margin of more than 7 aeIOU pts. Overall, our work establishes a new state of the art for time-interval prediction in TKGC.

## 6 Analysis

We investigate several research questions to gain further insights into our model:

**Q1.** What is the effect of each component of NeuSTIP on link and time-interval prediction?

**Q2.** How does our model perform in the inductive learning setting where the train and the test entities are disjoint?

**Q3.** How does NeuSTIP perform when there is limited training data available?

**Q4.** Are the rules generated by our model human-interpretable?

**Q5.** How important is it to consider all the 13 Allen predicates in our proposed model?

**Q6.** What is the effect of rule length on model performance?

For these analysis questions, unless otherwise stated, we use NeuSTIP (base) model – this allows us to directly assess the impact of design choices made in our model.

**Ablation study:** In order to understand the incremental contribution of each component (**Q1**), we conduct an ablation study by removing three components from NeuSTIP: Allen predicates (TR), PCA score (PCA) (Equation 4) and duration distribution (INTV) (Section 4.4). Please note that duration distribution is employed only during time interval prediction, so it will not change the results of link prediction. Likewise, Allen predicates (TR) are critical in predicting potential time intervals in our algorithm (Section 4.2.2) and cannot be absent from the model while doing time prediction.

Table 3 compares the link prediction performance of NeuSTIP model ablations. We observe that the presence of Allen relation constraints is critical for our model – it aids performance substantially in both datasets. We hypothesize that without them, low precision rules get too many groundings, which confuse the model. Next, the absence of PCA score (NeuSTIP - PCA) also hurts performance, as PCA acts as informed prior to the rule score.

Table 4 shows the effect of removing the duration distribution (INTV) and PCA score (PCA) on the time interval prediction of the model. We observe that both components help the model, although the contributions are not huge. See Appendix K for more details on ablation study.

**Inductive Setting:** It is generally believed that NS models perform well in inductive settings. To verify this for NeuSTIP (**Q2**), we conduct an experiment where the training and testing entities are

Table 4: Interval prediction performance of NeuSTIP and its variants, NeuSTIP-INTV (w/o interval distribution), NeuSTIP-PCA (w/o PCA score)

| Model | WIKIDATA12k aeIOU | YAGO11k aeIOU |
|---|---|---|
| NeuSTIP | **27.36** | **22.49** |
| NeuSTIP- INTV | 26.80 | 20.24 |
| NeuSTIP- PCA | 26.91 | 20.63 |

Table 5: Link prediction results in Inductive setting

| Model | WIKIDATA12k MRR | WIKIDATA12k H@1 | WIKIDATA12k H@10 | YAGO11k MRR | YAGO11k H@1 | YAGO11k H@10 |
|---|---|---|---|---|---|---|
| NeuSTIP | **25.30** | **16.03** | **44.46** | **20.01** | **14.10** | **31.31** |
| TLogic | 16.05 | 10.86 | 26.76 | 7.68 | 5.25 | 13.20 |

Table 6: Interval prediction results in Inductive setting

| Model | WIKIDATA12k aeIOU | YAGO11k aeIOU |
|---|---|---|
| NeuSTIP | **25.96** | **23.47** |
| Timeplex | 2.68 | 0.32 |

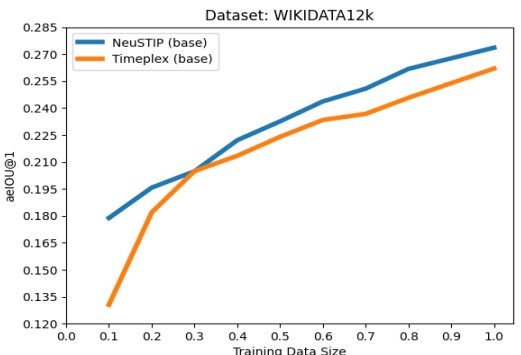

Figure 3: Limited training data experiment on WIKIDATA12k

Table 7: Link prediction and Time prediction performance on All vs integrated (ITG) Allen Predicates

| Model | WIKIDATA12k MRR | WIKIDATA12k H@1 | WIKIDATA12k H@10 | WIKIDATA12k aeIOU | YAGO11k MRR | YAGO11k H@1 | YAGO11k H@10 | YAGO11k aeIOU |
|---|---|---|---|---|---|---|---|---|
| NeuSTIP | **31.98** | **22.11** | **50.31** | **27.36** | **23.81** | **17.26** | **35.15** | **22.49** |
| NeuSTIP (ITG) | 29.49 | 18.78 | 50.18 | 23.37 | 20.18 | 13.87 | 30.59 | 16.06 |

disjoint. In order to generate the data for this experiment, we randomly select a subset of the temporal facts from the test data. Then, we remove any such temporal facts from the train data, which share a common entity with these test temporal facts. (Appendix L). We perform link and time interval prediction on the newly generated data in Table 5 and 6. We utilize TLogic and Timeplex as baselines for link and time interval prediction respectively. It can be clearly seen that our model outperforms Timeplex by a significant margin on aeIOU, because embedding-based models generally struggle in the inductive setting. Further, NeuSTIP outperforms TLogic on all metrics for link prediction supporting the hypothesis that NeuSTIP can generalize well to new data.

**Limited training data:** NS models have another advantage that they can learn in the presence of less amount of training data (**Q3**). We test this in Figure 3 on WIKIDATA12k dataset where we compare NeuSTIP (base) against Timeplex (base) on test data while training the models at varied training data sizes. We observe that when limited data is available (e.g. 10% data) NeuSTIP's rules are still able to capture the patterns in the data, while Timeplex struggles to perform well (Appendix M).

**Human interpretability of logical rules:** One advantage of our temporal rule-based model is that the predictions are in a human-interpretable form (**Q4**), whereas the predictions of embedding-based

models are opaque in nature. Here, we illustrate one real example from the YAGO11k dataset – it shows the reasoning behind predicting the correct entity/time interval by the rules of NeuSTIP model:

**Query:** (`Franz Dahlem`, `isAffiliatedTo`, ?, [`1920`, `1946`])

**Correct Answer:** `Communist Party of Germany`

**The rule that grounds the gold object:**

`isAffiliatedTo`(E1, E2, T1) ← `During`(T1, T2) ∧ `isMarriedTo`(E1, E3, T2) ∧ `Contains`(T2, T3) ∧ `isAffiliatedTo`(E3, E2, T3)

**The groundings:** E1: `Franz Dahlem`, E2: `Communist Party of Germany`, E3: `Kathe Dahlem`, T1: [`1920,1946`], T2: [`1899, 1974`], T3: [`1920,1946`]

The above rule provides an *explanation* of why `Franz Dahlem` was affiliated to `Communist Party of Germany` at a given time interval by reasoning that his wife `Kathe Dahlem` was also affiliated to the party during their marriage. (See Appendix N).

**Importance of All Allen predicates:** Recall that TILP is a recent model, which only uses three Allen relations in temporal rules: 'before', 'after' and a new aggregate relation 'touching', which combines all other Allen relation into one. In **Q5**, we wish to probe the value of considering all 13 Allen relations in NeuSTIP's rules, and compare it to a NeuSTIP version which uses the exact three relations used in TILP. We call this model NeuSTIP (ITG). The comparisons for link and time prediction of NeuSTIP and NeuSTIP (ITG) are provided in Table 7.

Table 8: Effect of rule length on link and time interval prediction for YAGO11k and WIKIDATA12k. Max RL denotes maximum **R**ule **L**ength that was considered.

| YAGO11k | | | | |
|---|---|---|---|---|
| **Max RL** | **MRR** | **Hits@1** | **Hits@10** | **aeIOU** |
| 3 | 23.81 | 17.26 | 35.15 | 22.49 |
| 2 | 10.95 | 10.43 | 11.65 | 19.23 |
| 1 | 9.39 | 9.17 | 9.58 | 18.37 |
| WIKIDATA12k | | | | |
| **Max RL** | **MRR** | **Hits@1** | **Hits@10** | **aeIOU** |
| 3 | 31.98 | 22.11 | 50.31 | 27.36 |
| 2 | 23.43 | 17.26 | 32.96 | 21.64 |
| 1 | 23.26 | 17.06 | 32.83 | 21.55 |

We observe that integrating many Allen relations into one has a profound impact on H@1 for both datasets, as the rules lose their preciseness. They become more generic and cause other entities to get many groundings, confusing the model. We further notice that integrating Allen predicates has far-reaching consequences on time prediction because Allen predicates play a crucial role in deciding and scoring the start and end time of time intervals in NeuSTIP. On further looking at data, we notice that Allen predicates that yield deterministic start/end points, such as `equals`, `finishes`, are quite important for good performance in time interval prediction. They should not be aggregated together, as that makes the scores assigned to start/end instances get distributed over a larger range, which affects the model performance.

**Rule Length:** We perform experiments to study the effect of rule length on the performance of NeuSTIP (**Q6**). We define a rule of rule length $i$ as one whose rule body consists of $i$ *KG relations* and $i$ *Allen relations*. A rule set of maximum (max.) rule length $i$ consists of all rules that have length from 1 to $i$ in it. For efficiency purposes, we restrict the maximum rule length to 3. Our results for link and time interval prediction are presented in Table 8.

From the table, we conclude that the rules of length 3 perform significantly better than the rules of length 2 because length 2 rules fail to capture the entity chains ($A \rightarrow D \rightarrow C \rightarrow B$) in cyclic rules such as $r_h(A,B,T1) \leftarrow A1(T1,T2), r(A,D,T2), A2(T2,T3), r^{-1}(D,C,T3), A3(T3,T4), r(C,B,T4)$ in its body which are important while finding the path from the head to the tail of a temporal fact.

To analyse the model further, we performed additional experiments on the effect of NeuSTIP performance on different types of temporal relations – instant relations (duration = 0), short relations (du-

ration $\leq 5$), and long relations (duration $> 5$). We find NeuSTIP outperforms baselines in all but one case (instant relations) reiterating the importance of finding groundings (Appendix P) in a rule-based model. Further details are in Appendix O.

## 7 Conclusions

In this paper, we develop a novel neuro-symbolic TKGC method that represents the temporal information of TKGs in a temporal rule language it defines, and uses rule groundings to ascribe a confidence score to a candidate answer, which can be an entity or a time interval. For that, it defines a novel scoring function. And, it also uses novel loss formulation for training. The key novelty of the proposed formulation is that it can perform both link prediction and time interval predictions in the neuro-symbolic setting. Compared to previous methods, our model has made substantial improvements in both link prediction and time interval prediction performance over two benchmark datasets. Furthermore, we show that our model is generalizable to new entities, is human-interpretable, and works well with limited data while answering a given link prediction and time prediction query. We release our code [2] and datasets for further research.

In the future, we will explore combinations of NeuSTIP with other orthogonal paradigms such as models that learn box embeddings instead of vector embeddings (Cai et al., 2021), and models based on graph neural network (Zhu et al., 2021). Exploring different methods of ensembling our rule-based and embedding-based models (apart from linear ensembling) can be another future work direction.

## Acknowledgements

This work is supported by IBM AI Horizons Network, grants from Google, Verisk, and Huawei, and the Jai Gupta chair fellowship by IIT Delhi. We also acknowledge travel support from the Professional Development Allowance (PDA) fund of IIT Delhi. We thank the IIT Delhi HPC facility for its computational resources. We sincerely thank Ananjan Nandi for useful discussions during the course of the research.

## Limitations

One limitation of our model, like rule-based models in general, is that they cannot directly capture the

---

[2] https://github.com/dair-iitd/NeuSTIP.git

numeric features present in the data. Although we exploited the gadgets subsumed in Timeplex (Jain et al., 2020) to capture numeric features present in TKGs, capturing these features directly inside our model is an interesting future direction. Further, our proposed model currently deals with closed path rules only and could benefit more if open paths can be exploited for time prediction.

## Ethics Statement

We anticipate no substantial ethical issues arising due to our work on link prediction and time interval prediction for Neuro-Symbolic TKGC.

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

## A  Allen's Interval Calculus

The temporal facts considered in our setting encode time intervals in the last argument, requiring the use of Allen relations as a formal technique that captures the temporal relations between the time intervals present in the data. As discussed in Section 3.1, we utilize all the 13 Allen relations possible between any two time intervals in our temporal rules. In this section, we describe all 13 Allen relations in Figure 4 in detail. Each of the relations in Allen algebra calculus can be written as set of rules. For example if we have one time interval $X = [X_{start}, X_{end}]$ and another time interval $Y = [Y_{start}, Y_{end}]$ then the Allen relation `before` exists between them, i.e. `before(X,Y)` iff $X_{start} < X_{end} < Y_{start} < Y_{end}$. Similarly, the constraints for each of the 13 *Allen relations* is defined in the last column named 'Chronological Sequence' of Figure 4.

## B  The Details and an Example of Temporal Logic Rule Extraction

We elaborate on temporal logic rule extraction defined in Section 4.1 along with an example. In order to carry out *all-walks*, it expresses TKG as graph $\mathcal{G}_{AW}$ wherein each quadruple is expressed as $\mathtt{s} \xrightarrow{(\mathtt{r},\mathtt{T})} \mathtt{o}$ denoting an edge $(\mathtt{r},\mathtt{T})$ between entities $\mathtt{s}$ and $\mathtt{o}$ (see Figure 1 (left)). NeuSTIP mines *all* walks over $\mathcal{G}_{AW}$ in three steps: $(a)$ First, beginning at $s$, it performs time-agnostic walks of length $m$ on $\mathcal{G}_{AW}$ such that the final node of the walk is $o$ (Figure 1 (middle)). It then expresses these walks in logical form. At this stage, these *all* walks exclusively consist of *KG relations*. $(b)$ Next, it introduces *Allen relations* into the *all-walks* captured thus far, in order to bind the time intervals existing between neighboring *KG relations* in the walk. Further, a special *Allen relation* is introduced to bind the time interval of target temporal fact and the first *KG relation* in the walk. $(c)$ At the final step, it substitutes the constants with variables to generalize grounded rule into a final rule (Fig. 1(right)).

Our example is based upon a fragment of the YAGO11k dataset which is shown in Figure 1. In this said example, `pF, pF`$^{-1}$`, wBi, iMT` denote abbreviations for the relations `playsFor`, `playFor`$^{-1}$`, wasBornIn, isMarriedTo` respectively. The relations `A1` to `A5` represent Allen relations. In order to learn a rule that is based upon target temporal fact `wBi(David, London, T6)` the model would first obtain a walk `David` $\xrightarrow{\texttt{(iMt,T4)}}$ `Victoria` $\xrightarrow{\texttt{(wBi,T5)}}$ `London` on $\mathcal{G}_{AW}$. This walk would further be expressed in the logical form as `iMt(David, Victoria, T4)`$\wedge$ `wBi(Victoria, London, T5)`.

In the next step (b), when *Allen relations* are introduced into the walk, the corresponding example would be expressed as `A4(T6,T4)` $\wedge$ `iMt(David, Victoria, T4)` $\wedge$ `A5(T4 ,T5)` $\wedge$ `wBi(Victoria, London, T5)`. Please note that these *Allen relations* denote one of the 13 relations in Allen Algebra calculus. The final rule after introducing variables is `wBi(A, B, C)` $\leftarrow$ `A4(C, F)` $\wedge$ `iMt(A, D, F)` $\wedge$ `A5(F, G)` $\wedge$ `wBi(D, B, G)`. This rule is expressed without the entity and time interval variables in Figure 1(right) as `wBi` $\leftarrow$ `A4` $\wedge$ `iMt` $\wedge$ `A5` $\wedge$ `wBi`.

## C  GRU for Embedding-based Rule Score

In this section, we provide the details of GRU utilized in Section 4.2.1. NeuSTIP learns a unique embedding $\mathbf{L_j}$ representation for the body $\wedge_{t=1}^m A_t \wedge r_t$ of a given rule $L_j$. Motivated by ARLE-IR model (Mei et al., 2022), we employ Gated Recurrent Unit (GRU) model (Cho et al., 2014) to learn $\mathbf{L_j}$ embedding. At time $t$, the input of the form $\mathbf{x}_t = [\mathbf{A}_t; \mathbf{r}_t]$ is fed to GRU where $\mathbf{A}_t$ and $\mathbf{r}_t$ are the embedding vectors of t-th *Allen predicate* $A_t$ and *KG relation* $r_t$ in a rule $L_j$'s body. GRU unit utilizes the following functions in order to generate the hidden-layer embedding $\mathbf{h}_t$ at time $t$:

$$\mathbf{r}_t = \sigma\big(\mathbf{W}_r \cdot \mathbf{x}_t + \mathbf{U}_r \cdot \mathbf{h}_{t-1} + \mathbf{b}_r\big) \qquad (9)$$

$$\mathbf{z}_t = \sigma\big(\mathbf{W}_z \cdot \mathbf{x}_t + \mathbf{U}_z \cdot \mathbf{h}_{t-1} + \mathbf{b}_z\big) \qquad (10)$$

$$\mathbf{n}_t = \tanh\big(\mathbf{W}_n \cdot \mathbf{x}_t + \mathbf{r} \odot \mathbf{h}_{t-1} + \mathbf{b}_n\big) \quad (11)$$

$$\mathbf{h}_t = (1 - \mathbf{z}_t) \odot \mathbf{n}_t + \mathbf{z}_t \odot \mathbf{h}_{t-1} \qquad (12)$$

where $\mathbf{r}_t$ is the *reset* gate that allows the hidden state to discard information that is insignificant in the future and $\mathbf{z}_t$ is the *update* gate that controls how much information from $\mathbf{h}_{t-1}$ is carried over to $\mathbf{h}_t$. The final hidden state embedding $\mathbf{h}_m$ after $m$ sequential steps of GRU represents the path embedding $\mathbf{L_j}$ of a given rule $L_j$.

| Allen Predicates | | Pictoral Example | Chronological Sequence |
|---|---|---|---|
| Relations | Inverse Relations | | |
| before(X, Y) | after(Y,X) | X ——— Y — | $X_{start} < X_{end} < Y_{start} < Y_{end}$ |
| equals(X, Y) | equals(Y,X) | X ——— / Y ——— | $X_{start} = Y_{start} < X_{end} = Y_{end}$ |
| meets(X, Y) | met by(Y, X) | X ——— Y ——— | $X_{start} < X_{end} = Y_{start} < Y_{end}$ |
| overlaps(X, Y) | overlapped by(Y, X) | X ——— Y ——— | $X_{start} < Y_{start} < X_{end} < Y_{end}$ |
| contains(X, Y) | during(Y,X) | X ——— Y — | $X_{start} < Y_{start} < Y_{end} < X_{end}$ |
| starts(X, Y) | started by(Y, X) | X — Y ——— | $X_{start} = Y_{start} < X_{end} < Y_{end}$ |
| finishes(X, Y) | finished by(Y, X) | X — Y ——— | $Y_{start} < X_{start} < X_{end} = Y_{end}$ |

Figure 4: This figure lists all the 13 relations in Allen algebra calculus (Allen, 1983). The pictorial example in the third column is for the relations in the first column

## D  PCA Score Metric for Temporal Data

Here we explain PCA score utilized in Section 4.2.1. PCA metric (Galárraga et al., 2013) is based on the Partial Closed World assumption according to which if we know one object $\mathtt{o}$ for a given $\mathtt{s}$ and $\mathtt{T}$ in a temporal fact $(\mathtt{s}, \mathtt{r_h}, \mathtt{o}, \mathtt{T})$ then we know all the $\mathtt{o}'$ for that $\mathtt{s}$ and $\mathtt{T}$. If we consider temporal rule $L_j$ to be $\mathtt{B} \Rightarrow \mathtt{r_h}(\mathtt{s}, \mathtt{o}, \mathtt{T})$, the PCA score of this rule for link prediction, $\psi_{stat}(L_j)[o]$, is:

$$\frac{\#(\mathtt{s}, \mathtt{o}, \mathtt{T}) : |\mathtt{N}(\mathtt{s}, \mathtt{B}, \mathtt{o}, \mathtt{T})| > 0 \wedge \mathtt{r_h}(\mathtt{s}, \mathtt{o}, \mathtt{T}) \in \mathtt{P}}{\#(\mathtt{s}, \mathtt{o}, \mathtt{T}) : |\mathtt{N}(\mathtt{s}, \mathtt{B}, \mathtt{o}, \mathtt{T})| > 0 \wedge \exists \mathtt{o}' : \mathtt{r_h}(\mathtt{s}, \mathtt{o}', \mathtt{T}) \in \mathtt{P}}$$

Here, $\mathtt{N}(\mathtt{s}, \mathtt{B}, \mathtt{o}, \mathtt{T})$ denotes the path in the body $\mathtt{B}$ of the rule $L_j$. This implies that we divide the number of positive examples $\mathtt{P}$ satisfied by the rule by the total number of $(\mathtt{s}, \mathtt{o}, \mathtt{T})$ satisfied by the rule such that $\mathtt{r_h}(\mathtt{s}, \mathtt{o}', \mathtt{T})$ is a positive example for some $\mathtt{o}'$. Similarly, we define the PCA score for start time instance $t_b$, $\psi_{stat}(L_j)[t_b]$, as

$$\frac{\#(\mathtt{s}, \mathtt{o}, \mathtt{t_b}) : |\mathtt{N_{tb}}| > 0 \wedge \exists \mathtt{t_e} \in \mathtt{T}, \mathtt{r_h}(\mathtt{s}, \mathtt{o}, [\mathtt{t_b}, \mathtt{t_e}]) \in \mathtt{P}}{\#(\mathtt{s}, \mathtt{o}, \mathtt{t_b}) : |\mathtt{N_{tb}}| > 0 \wedge \exists \mathtt{T}' : \mathtt{r_h}(\mathtt{s}, \mathtt{o}, \mathtt{T}') \in \mathtt{P}}$$

$|\mathtt{N_{tb}}|$ is a notation for $|\mathtt{N}(\mathtt{s}, \mathtt{B}, \mathtt{o}, \mathtt{t_b})|$. This implies that we divide the number of positive examples $\mathtt{P}$ satisfied by the rule by the total number of $(\mathtt{s}, \mathtt{o}, \mathtt{T})$ satisfied by the rule such that $\mathtt{r_h}(\mathtt{s}, \mathtt{o}, \mathtt{T}')$ is a positive example for some $\mathtt{T}'$.

## E  Distance Computation between Time Instances

In order to find the distance $\mathtt{d}$ between two time instances $\mathtt{t_a}$ and $\mathtt{t_b}$ i.e. $\mathtt{d}(\mathtt{t_a} - \mathtt{t_b})$, in Section 4.3 the model sorts the years in $\mathcal{T}$ in increasing order and assign a unique id to each of them. The difference $\mathtt{d}$ is then taken between those ids. The difference is then divided by the maximum difference between any two ids, in order to follow the constraint that $0 \leq \mathtt{d}(.) \leq 1$.

## F  Data Statistics

The details of the datasets used for experimentation in Section 5 are provided in Table 9. We utilize two standard TKG datasets - YAGO11k and WIKI-DATA12k for our experimentation. Both these datasets are time interval-based datasets. We utilize the standard train, valid and test splits for these datasets in our experiments.

Table 9: Statistics of Temporal KG datasets

| Features | YAGO11k | WIKIDATA12k |
|---|---|---|
| #Entities | 10622 | 12554 |
| #Relations | 10 | 24 |
| #Instants | 251 | 237 |
| #Intervals | 6651 | 2564 |
| #Training | 16408 | 32497 |
| #Validation | 2051 | 4062 |
| #Test | 2050 | 4062 |

## G  Experimental Details for NeuSTIP

For all the results reported for the proposed model in Table 1 and 2, we optimize parameters of the loss function defined in Section 4.3 with an Adam optimizer (Kingma and Ba, 2015) while decreasing

the learning rate by Cosine Annealing ensuring that the minimum learning rate at any time during the training does not fall below the one-fifth of its initial value. To get the best results for link prediction, we train our model for 5000 epochs. Likewise, for time interval prediction, we train the model for 2000 epochs. We set a dimensionality for all the *Allen relation* embeddings, *KG relation* embeddings, and the rule head embeddings (which have the same dimension as the hidden dimension of the GRU) to be 32. Besides, we set the maximum rule length as 3 for both datasets.

Further, $\eta$ in Equation 8 is a learnable parameter that is trained along with the rule score $\psi(L_j)[c]$ in Equation 4. The initial value of $\eta$ is chosen as a hyperparameter and is tuned over the dev set selecting the best value of MRR for link prediction and aeIOU for time prediction.

During the training of the model, we select the best validation model for link prediction based on the MRR metric and the best validation model for time interval prediction based on the aeIOU metric. Further details of the hyperparameters adopted for all the experiments are provided in Appendix H and more details about restrictions on the ruleset are explained in Appendix I.

## H  Hyper-Parameter Settings for NeuSTIP

For both link and time interval prediction in Table 1 and 2, we set our learning rate to be $1e-3$ for both datasets. Table 10 lists the values of the coefficient $\eta$ (in Equation 8) which we multiply to the overall KGE score while ensembling with our rule-based model for the two datasets. For link prediction, we consider $\eta$ from the set {0, 1e-3, 1e-2, 1}, and select the best value of $\eta$ based on MRR on the dev set. For time interval prediction, we consider $\eta$ from the set {0, 1e-3, 1e-2, 1e-1, 1} and determine the best value of $\eta$ based on aeIOU on the validation set.

Table 10: The best hyperparameter settings for link and time prediction over two TKG datasets

| Hyper-parameter | YAGO11k | WIKIDATA12k |
|---|---|---|
| $\eta$ (Link prediction) | 1e-3 | 1e-2 |
| $\eta$ (Time prediction) | 1e-1 | 1e-1 |

## I  More Implementation Details

During the process of rule extraction in Section (4.1), given the temporal fact $\mathtt{r_h(s,o,T)}$ in the rule head, we disallow this temporal fact to re-occur in the body of the rule to avoid mining cyclic rules. Further, there are some cases in time prediction where we have 0 groundings for all the instances with respect to both start and end scores. In such cases, for a given relation $\mathtt{r}$, we predict *t_start* as *mean_start[r]*, and *t_end* as *mean_start[r] + mean_offset[r]*. Here, *mean_start[r]* and *mean_offset[r]* are the average start and the average offset of intervals for the relation $\mathtt{r}$, computed in terms of the assigned ids (as explained in Appendix E).

While computing $\phi(path)$ (Eqn. 5) for the start and end time instances in time interval prediction, there are also cases when $r_h = r_1$ (see Eqn 1). For such cases, the Gaussian distributions are computed for the difference between successive occurrences of start/end time for a fixed $(s, r)$. In such cases, given the fact $r_1(e_1, e_2, [t2_b, t2_e])$, NeuSTIP calculates path score for start time candidate $t1_b$ in query $r_h(e_1, e_{m+1}, ?)$ as,

$$\phi(path)[t1_b] = \mathcal{N}(abs(t1_b - t2_b)|\mu_{r_h r_1}^B, \sigma_{r_h r_1}^B)$$
(13)

Here, $abs$ represents absolute value. The same approach is used to estimate $\phi(path)[t1_e]$.

Additionally, the loss function for Time Interval prediction (see Section 4.3) is normalized for each positive start instance by dividing it by $\sum_{n_b \in \mathbf{N}_b} d(n_b, t_b)$ for a given $t_b$, and similarly for each positive end instance.

## J  Statistical Richness of the Rules

We compute metrics such as the number of rules, and the average number of groundings per rule as a measure for computing the statistical richness of the rules obtained in our model. The proposed methodology of performing all walks in NeuSTIP generated 8186 rules for YAGO11k and 31,807 rules for WIKIDATA12k dataset. The reason for obtaining a moderate number of rules while we consider all walks on TKGC is the sparsity of temporal KGCs as already discussed in Section 4.1. Further, the average number of groundings per rule considering the train set is 17.87 for YAGO11k, and 885.21 for WIKIDATA12k, which is sufficiently high.

## K  Link Prediction for Ablation Study

This section provides the details of the experimental setup for the ablation study on link prediction

performed in Table 3 in Section 6 (**Q1**). In order to perform link prediction for the models: NeuSTIP, NeuSTIP- TR and NeuSTIP-PCA, we utilize the same initial value of hyperparameters as reported in Table 10 in Appendix H. To train the model with rules that are without the Allen relations (NeuSTIP-TR), we essentially treat the absence of Allen relations as a special 'NOREL' constraint, and we feed the embedding of this constraint as input $x_t$ (Appendix C) to the GRU at every step instead of giving the Allen relation embeddings as input.

## L Experimental Setup for Inductive Learning Study

This section explains the experimental setup for the inductive learning study (**Q2**) performed in Section 6. Our experimental setup is motivated by a similar study performed in DRUM model (Sadeghian et al., 2019).

**Data generation:** In order to generate our *inductive test data*, we randomly select a subset of the temporal facts from the standard test data. In our experiments, we chose 50% of the standard test data. We train our model on one set of *inductive training data* that helps the model in learning rules along with their confidence. It is to be noticed that if we utilize the *inductive training data* as background knowledge at the test time, then because of the disjoint set of entities at the train and the test time, we would not be able to obtain any groundings of the rules. To overcome this, we employ the standard train data and validation data as the background knowledge at the test time in order to ground the body of the rules. The key point here is that the rules along with their confidences that are learned on one set of entities can be applied to a different set of entities as well since the rules are *generalizable* and are not bound to fixed entities.

We compared NeuSTIP with TLOGIC (Liu et al., 2021) which addresses the problem of link forecasting on temporal knowledge graphs. Unlike our solution, TLOGIC works with the time-instant dataset. Therefore, for a comparable setup, we used our datasets only while treating the start time instant of the interval as the timestamp of the respective fact. We used the publicly available [3] TLOGIC implementation and ran it using the same training data and background knowledge as that for NeuSTIP. The ablation study of TLOGIC suggested that the higher value of the hyperparameters,

---

[3] https://github.com/liu-yushan/TLogic

like the number of random walks and time window size, leads to better model performance. Therefore, for both datasets, we increased the number of random walks to $20K$ from its default value of 200 and set the time window size to 1000. Similar to NeuSTIP, we generated rules of lengths 1,2, and 3 using the exponential transition distribution of TLOGIC that performs better than the uniform distribution.

## M Limited Data Study

This section is complementary to the limited data study conducted in Section 6 (**Q3**) as we perform a limited data study for the YAGO11k dataset here. We consider different percentages of the original data (10% to 100%) and train NeuSTIP (base) on this data and test the model using the standard test data. We compare our model against the Timeplex (base) model in Figure 5. As can be seen, in limited training data areas (10% of total training data), the gap between the time-prediction performance of our model and Timeplex becomes quite significant. While embedding-based models are expected to not perform well in the data-scarce scenario, the strategy of NeuSTIP to handle the cases where no relevant rule is grounded helps in improving the model performance significantly. Such a scenario is often encountered in the limited data setting. To this end, when no rule is grounded for query $r(s, o, ?)$, NeuSTIP uses a priori knowledge of the mean of start time instants $t_b^{avg}$ and the average duration $(intv)$ of relation $r$ and to predict the time interval as $[t_b^{avg}, t_b^{avg} + intv]$.

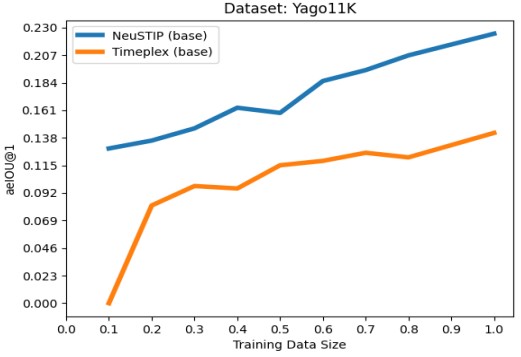

Figure 5: Limited training data experiment on YAGO11k

## N Human interpretability of Rules

This section provides more examples of the interpretability feature (**Q4**) of NeuSTIP put forward in

Section 6. Next, we provide another example of an interpretable rule from the YAGO11k dataset for time prediction as below.

**Query:** (`Donna Hanover, isMarriedTo, Rudy Giuliani, ?`)
**Correct Answer:** [1984, 2002]
**Rule grounding gold start and gold end:**
$\texttt{isMarriedTo}(E1, E2, T1) \leftarrow \texttt{Equals}(T1, T2) \wedge \texttt{isMarriedTo}^{-1}(E1, E2, T2)$
**The Grounding:** E1: `Donna Hanover`, E2: `Rudy Giuliani`, T2: [1984,2002]

The above rule provides the *temporal information* that since `Rudy` was married to `Donna` for a given time interval, `Donna` was also married to `Rudy` for the exact same time interval due to the symmetric nature of the `isMarriedTo` relation.

Next, we present two more examples from WIKI-DATA12k dataset on how the rules of NeuSTIP provide human interpretable predictions. We begin by considering an example of link prediction in WIKIDATA12k dataset.

**Query:** (`Ammerschwihr, liate, ?, [1920, present]`)
**Correct Answer:** `Haut-Rhin`
**The rule that grounds the gold object:**
$\texttt{liate}(E1, E2, T1) \leftarrow \texttt{MetBy}(T1, T2) \wedge \texttt{liate}(E1, E3, T2) \wedge \texttt{Equals}(T2, T3) \wedge \texttt{liate}^{-1}(E3, E4, T3) \wedge \texttt{Meets}(T3, T4) \wedge \texttt{liate}(E4, E2, T4)$
**Groundings:** E1: `Ammerschwihr`, E2: `Haut-Rhin`, E3: `Upper Alsace`, E4: `Soultzmatt`, T1: [1920, present], T2: [1871, 1920], T3: [1871, 1920], T4: [1920, present]

Here, the abbreviation `liate` in WIKIDATA12k stands for `located in the administrative territorial entity`. The above rule provides an *explanation* of why `Ammerschwihr` is located in the administrative entity of `Haut-Rhin` since 1920, by reasoning that `Ammerschwihr` and `Soultzmatt` were both a part of the entity `Upper Alsace` for the same time interval just before 1920, and `Soultzmatt` became a part of `Haut-Rhin` in year 1920. The next example explains time prediction in WIKIDATA12k by NeuSTIP ruleset.

**Query:** (`Turku, country, Russian Empire, ?`)
**Correct Answer:** [1809,1917]
**Rule that grounds the gold start and gold end:**
$\texttt{country}(E1, E2, T1) \leftarrow \texttt{Meets}(T1, T2) \wedge \texttt{country}(E1, E3, T2) \wedge \texttt{Equals}(T2, T3) \wedge \texttt{country}^{-1}(E3, E4, T3) \wedge \texttt{MetBy}(T3, T4) \wedge \texttt{country}(E4, E2, T4)$
**Rule Grounding:** E1: `Turku`, E2: `Russian Empire`, E3: `Finland`, E4: `Mikkeli Province`, T2: [1917,present], T3: [1917,present], T4: [0,1917]

The above rule *explains* that `Turku` is a part of the country `Finland` since 1917, and another place `Mikkeli Province` is also a part of `Finland` since 1917. Just before this, `Mikkeli Province` was a part of `Russian Empire`, so the rule provides us the *temporal information* that `Turku` was also a part of `Russian Empire` just before being a part of `Finland`.

## O Time Interval Prediction for Distinct Relation Classes

Table 12: aeIOU across relation classes in YAGO11k and WIKIDATA12k datasets

| YAGO11k | | | |
|---|---|---|---|
| **Model** | **Instant** | **Short** | **Long** |
| NeuSTIP (base) | 9.32 | 23.93 | 58.71 |
| NeuSTIP w/ KGE | 19.03 | 24.59 | 60.43 |
| Timeplex | 18.39 | 20.63 | 24.80 |
| **WIKIDATA12k** | | | |
| **Model** | **Instant** | **Short** | **Long** |
| NeuSTIP (base) | 35.44 | 25.98 | 22.78 |
| NeuSTIP w/ KGE | 36.97 | 26.11 | 27.49 |
| Timeplex | 34.51 | 24.80 | 20.93 |

The goal of this experiment is to observe the performance of the proposed model for different classes of relations present in the TKG. Motivated by Timeplex model (Jain et al., 2020), we categorize the relations into three classes: `Instant`, `Short` and `Long`. `Instant` relations are those whose start time and the end time coincide in a given time interval. Each relation in the `Short` category has an average time duration of less than five years. Each relation in the `Long` category has an average time duration greater than five years. A category-wise detail of relations in YAGO11k and WIKIDATA12k datasets is shown in Table 11.

The detailed results of NeuSTIP (base), NeuSTIP w/ KGE, and Timeplex across different relation classes are presented in Table 12. As we observe NeuSTIP (base) considerably outperforms Timeplex in five out of six cases. This supports the hypothesis that our proposed model captured the patterns present in different relation classes by its learned rules. Further, NeuSTIP w/ KGE consistently outperforms the Timeplex model. Please note that the reason for the performance degradation of NeuSTIP (base) for `Instant` category in YAGO11K is that there are no groundings available

Table 11: Categorization of YAGO11k and WIKIDATA12k relations into `Instant`, `Short` and `Long` categories.

| YAGO11k Relation-wise Category | | |
|---|---|---|
| **Instant** | **Short** | **Long** |
| diedIn, WasBornIn | playsFor, graduatedFrom | isMarriedTo, isAffiliatedTo, owns, worksAt, created, hasWonPrize |
| **WIKIDATA12k Relation-wise Category** | | |
| **Instant** | **Short** | **Long** |
| nominatedFor, winner, academic − Degree | memberOfSportsTeam, award −Received, significantEvent, twinnedAdministrativeBody, educatedAt | positionHeld, locatedInAdministrativeTerritorialEntity, spouse instanceOf, employer, memberOf, countryOfCitizenship, titleOfChessPerson, memberOfPoliticalParty, residence, country containsAdministrativeTerritorialEntity, capitalOf, IMAStatsAndOrRank, heritageDesignation, headOfGovernment |

for many temporal facts lying in this category. Accordingly, the explanation for the high performance in `Long` category of YAGO11k is that the model discovers perfect rules for a considerable amount of temporal facts which results in a performance boost. For instance, for the relation `isMarriedTo` that belongs to the `Long` category, the model discovers the inverse of `isMarriedTo` in the rule set and yields better performance by exploiting the symmetric nature of this relation. The other `Long` relations where we perform better than TimePlex are `hasWonPrize` and `isAffiliatedTo`. In these relations, there are a substantial number of temporal facts for which closed walk groundings exist, and hence the target start time and end time are often grounded in these cases, leading to better performance.

## P  Effect of Groundings on the Model Performance

This empirical study is motivated by the question: how critical are the rule groundings in order to achieve superior performance in the proposed model? In order to conduct this study, we train the NeuSTIP (base) in a typical setting. However, at the test time, we categorize our test temporal facts into two categories: **GR** and **NGR**. For a link prediction query (s,r,?,T), **GR** encodes those temporal facts for which the true tail answer 'o' is reached by following the grounded path of at least one rule's body while beginning the path at the temporal fact head 's', otherwise the temporal fact is assigned to **NGR** category. Likewise, for a time prediction query r(s,o,?), **GR** encodes those temporal facts for which the true start time '$t_b$' and the true end time '$t_e$' are grounded by some rule. Specifically, if we

obtain a positive score in Equation 5 for the true start time and true end time, then we will assign the temporal fact to **GR**, otherwise we assign it to **NGR**. Here, of course, the path also has to be from the head 's' to the tail 'o' in the rule body. The effect of rule groundings on link prediction and time interval prediction are presented in Table 13.

Table 13: Effect of rule groundings on link and time prediction for YAGO11k and WIKIDATA12k datasets.

| YAGO11k | | | | |
|---|---|---|---|---|
| **Category** | **MRR** | **Hits@1** | **Hits@10** | **aeIOU** |
| **GR** | 33.57 | 24.48 | 49.36 | 30.07 |
| **NGR** | 0.018 | 0.0 | 0.0 | 9.86 |
| **WIKIDATA12k** | | | | |
| **Category** | **MRR** | **Hits@1** | **Hits@10** | **aeIOU** |
| **GR** | 36.03 | 24.55 | 50.31 | 29.65 |
| **NGR** | 0.016 | 0.0 | 0.0 | 14.89 |

As can be observed from the table, the performance of the model gets disappointingly low when the model can not discover the groundings of a target tail in a given rule set (**NGR** class). Based on this study, we conclude that the performance of the model critically depends upon whether a path exists from the head to tail in a given TKG in order to generate non-zero scores for a temporal fact. This is typical behavior of NS-KGC models and similar behavior has been observed in the past for static NS-KGC models such as RNNLogic (Qu et al., 2021), ExpressGNN (Zhang et al., 2020) models. However, this effect can be alleviated to some extent by rule augmentation techniques recently proposed in Nandi et al. (2023). Contrastingly, the TKGE models would always generate a non-zero score for a given temporal fact because they employ a scoring function that composes the embeddings to generate the resulting score without fail.

## Q Effect of the first Allen Relation on Model Performance

Table 14: NeuSTIP model's performance based on first Allen relation $a_1$ in rule language.

| YAGO11k | | |
|---|---|---|
| **Allen Relation** | **MRR** | **aeIOU** |
| DET | 18.62 | 22.55 |
| NONDET | 17.50 | 16.62 |
| OVERALL | 23.81 | 22.49 |
| **WIKIDATA12k** | | |
| **Allen Relation** | **MRR** | **aeIOU** |
| DET | 23.45 | 27.31 |
| NONDET | 25.30 | 21.97 |
| OVERALL | 31.98 | 27.36 |

It is very difficult to study the impact of Allen relations present in a given rule in our current rule language in general because multiple Allen relations are present in one rule. Further, Allen relations are interleaved between the KG relations to bind the time of two neighboring KG relations in the rule body. However, the first Allen relation $a_1$ present in rule body (Equation 1) has its unique contribution to the score computation, especially in the case of time interval prediction as discussed in the paper. Hence, for our study, we divide Allen predicates into two categories: DET refers to the Allen predicates that fix either the start or the end time of the time interval for e.g.: `finishedby`, `meets`, `metby`, `starts`, `startedby`, `equals` and NON-DET refers to the Allen predicates where both ends are free, for example: `before`, `after`, `overlaps`, `during`, `contains`, `overlappedby`.

We, then, categorize the rules into two categories based upon whether the first Allen relation of the rule falls into DET or NONDET category while ignoring the rules in the other category and predicting scores in Equation 2 based on rules in a given category. We also consider a third category - OVERALL - that constitutes the results in Table 1 and 2 of the paper that are obtained with all rules considered. The results for the link prediction and time interval prediction in three categories are presented in Table 14.

Results in DET category are better than NON-DET category for link prediction for YAGO11k dataset and the results of NONDET category are better than DET category for link prediction for WIKIDATA12k dataset. For time interval prediction, DET alone can help the model in achieving performance similar to OVERALL model performance for time prediction. Further, for link prediction, using only one of them reduces the performance as compared to the OVERALL mode. This is because the head Allen relation does not as directly influence the entity reached by the rule during link prediction as compared to the time prediction. The impact on link prediction seems to be governed by the total number of rules, as more rules help in grounding the gold entity because more paths are available to ground them. Because of this, when the rules are divided into DET and NONDET categories, fewer rules are available for link prediction, resulting in a drop in performance. In summary, using both sets of relations finds competitive performance in all settings.

## R Influence of Ensembling on the Proposed Model

Our goal here is to study the question: when we ensemble NeuSTIP with Timeplex in Table 1, how do they influence each other? In order to answer this, we considered the ranks of NeuSTIP (base), Timeplex and an ensemble (NeuSTIP w/ KGE) individually for each test quadruple obtained during link prediction and performed a comparative study to understand how the KGE models contribute to the performance of NeuSTIP.

Our major conclusion after analysis is that NeuSTIP mainly fails for the cases when no rule gets fired for a given quadruple for the gold entity resulting in a zero score in Equation 2 in paper. For instance, we found 796 such quadruples (out of a total 4102 test instances) in YAGO11k and 609 quadruples (out of 8124 quadruples in test data) in WIKIDATA12k, where our model generated zero score. For such instances, the average rank of our model was 5327 for YAGO11k, and 6307 for WIKIDATA12k while the average rank of Timeplex for such cases was 1654.51 for YAGO11k and 1420.07 for WIKIDATA12k. Therefore, our model relies on ensembling for the cases when it obtains no groundings for any rules learnt by NeuSTIP for a given quadruple. However, for the cases when the model obtains the groundings, it performs better than the corresponding Timeplex and Ensemble (NeuSTIP w/ KGE) model. For instance, when NeuSTIP obtained its groundings, its average rank is 51.1, while the average rank of Timeplex for such cases is 357.54, and is 169.70 for ensemble for YAGO11k datasets. Similarly, the average rank of NeuSTIP for cases when groundings were found is 38.09 while the average rank of Timeplex for such cases is 103.27, and ensemble for such cases

is 88.65 for WIKIDATA12k. Hence, we conclude that NeuSTIP relies on ensembling when it finds no grounding but outperforms the ensembling and the Timeplex model when it obtains the groundings for the candidate entity in link prediction for the rules learnt by NeuSTIP.

## S   Influence of Ensembling on Contemperaneous Models

The significant performance gain of NeuSTIP after ensembling it with Timeplex (a pure neural approach) directed us to study the impact of the ensembling approach on the other baseline models. To that end, we experimented (results in Table 15 below) with the state-of-the-art baseline model TILP (Xiong et al., 2023) ensembled with the Timeplex for different values of eta hyperparameter that influence the weightage of both approaches (see Equation 8). In our preliminary attempt at ensembling, we observed a different trend compared to what we noted in NeuSTIP, i.e., the ensembling did not improve the performance of TILP. This counter-intuitive observation points towards the sophisticated process of ensembling. An in-depth analysis of ensembling methods for leveraging the goodness of symbolic and neural approaches can be an interesting future research direction.

Table 15: Effect of Ensembling TILP with Timeplex.

| ETA | 10e-3 | 10e-5 | 10e-6 | 10e-8 | 0 |
|------|-------|-------|-------|-------|-------|
| H@1 | 10.09 | 11.16 | 11.26 | 11.80 | 16.81 |
| H@10 | 14.19 | 16.86 | 17.47 | 18.79 | 41.39 |
| MRR | 11.69 | 13.24 | 13.50 | 14.25 | 25.32 |

## T   Future Link Forecasting using NeuSTIP Model

Our goal here is to perform future link forecasting on NeuSTIP (base) such that time instances in train and test sets are chronologically ordered and are disjoint. We now explain the experimental setup followed by the results.

Table 16: Possible range of start time instances in the datasets for link forecasting

| YAGO11k | |
|---|---|
| Train start times range | [$t_{min}$, 2007] |
| Valid start times range | [2008, 2011] |
| Test start times range | [2012, $t_{max}$] |
| **WIKIDATA12k** | |
| Train start times range | [$t_{min}$, 2009] |
| Valid start times range | [2010, 2012] |
| Test start times range | [2013, $t_{max}$] |

**Experimental Setup of NeuSTIP (base):** We first sorted YAGO11k/WIKIDATA12k datasets according to the start time of time intervals, to make them suitable for forecasting ensuring the time in train and test sets are disjoint. Specifically, we list the possible values of the start time for train, validation, and test phases in YAGO11k/WIKIDATA12k datasets in the Table 16. The size of YAGO11k dataset (number of quadruples) after this split (without adding inverses) is: 16,806 (train), 1823 (valid), 1880 (test). Likewise, the size of WIKIDATA12k dataset after this split is : 33,368 (train), 3649 (valid), 3604 (test) without adding inverses. We then conducted our experiment with the following constraints: ($a$) during the rule generation phase, we ensure that the start time of any interval occurring in the body of the rule is strictly less than that of the quadruple used in the head. Consequently, our model finds fewer rules. ($b$) Similarly during grounding the rules, while performing parameter learning, we maintain the above condition. ($c$) While running inference on the test set, this condition is automatically ensured due to the way in which the dataset is split. It is important to note that due to fewer rules and lesser groundings found in this split, our model's performance also reduces.

**Results of NeuSTIP (base) for Link Forecasting:** we obtained the final results of link forecasting by employing time-aware filtering (Jain et al., 2020) that we have employed in all the experiments in our paper. The results are given in Table 17.

Table 17: Link Forecasting for NeuSTIP (Base)

| YAGO11k | | |
|---|---|---|
| **MRR** | **H@1** | **H@10** |
| 7.08 | 3.38 | 13.86 |
| **WIKIDATA12k** | | |
| **MRR** | **H@1** | **H@10** |
| 13.82 | 7.45 | 26.07 |

**Link Forecasting for TANGO model:** We now explain link forecasting in TANGO model (Han et al., 2021b) which is a past model in literature from which our link forecasting experiment has been motivated. Please note that the results obtained below are not directly comparable to the results of link forecasting in NeuSTIP (base) provided above because NeuSTIP is a time-interval dataset and TANGO is a time-instance dataset, their time-aware filtering techniques are different from each other. We are providing these results only for completeness.

Table 18: Results of TANGO model for link forecasting in YAGO11k and WIKIDATA12k datasets.

| YAGO11k | | | |
|---|---|---|---|
| **Model** | **MRR** | **H@1** | **H@10** |
| Raw | 2.073 | 0.638 | 4.681 |
| Time Aware filter | 2.089 | 0.665 | 4.681 |
| Time Unaware filter | 2.726 | 1.197 | 5.346 |
| **WIKIDATA12k** | | | |
| **Model** | **MRR** | **H@1** | **H@10** |
| Raw | 4.718 | 2.79 | 8.075 |
| Time Aware filter | 4.83 | 2.917 | 8.131 |
| Time Unaware filter | 6.39 | 4.735 | 8.963 |

**Experimental Setup of TANGO model:** We obtained results of link forecasting for the TANGO model on the YAGO11k/WIKIDATA12k datasets. We further $(a)$ consider the start time of all the quadruples, since the TANGO model is inherently designed for time instance datasets and cannot directly work with time-interval datasets as our model. $(b)$ We utilized the publicly available code[4], and ran it on the default set of hyperparameters after pre-processing the datasets in exactly the same way as the model does for ICEWS05-15 dataset.

Below, in Table 18, we provide the results of applying the TANGO model for link forecasting on YAGO11k/WIKIDATA12k datasets for different filtering settings. For the definition of the raw, time-aware, and time-unaware filtering please refer to TANGO model (Han et al., 2021b).

**NeuSTIP vs TANGO comparison:** Lastly, we obtained the results of NeuSTIP (Base) for link forecasting for raw (without filtering) settings so that the results of TANGO and NeuSTIP can be compared directly for raw settings in Table 19.

Table 19: Comparison of TANGO and NeuSTIP (base) model for link forecasting in YAGO11k and WIKIDATA12k datasets for raw filtering setting.

| YAGO11k | | | |
|---|---|---|---|
| **Model** | **MRR** | **H@1** | **H@10** |
| NeuSTIP (Base) | 5.289 | 0.0 | 12.978 |
| TANGO | 2.073 | 0.638 | 4.681 |
| **WIKIDATA12k** | | | |
| **Model** | **MRR** | **H@1** | **H@10** |
| NeuSTIP (Base) | 9.455 | 0.0 | 23.557 |
| TANGO | 4.718 | 2.79 | 8.075 |

As can be observed, our model outperforms TANGO for link forecasting on both YAGO11k and WIKIDATA12k on MRR and H@10.

---

[4] https://github.com/TemporalKGTeam/TANGO