# OpenReview forum: "NeuSTIP: A Neuro-Symbolic Model for Link and Time Prediction in Temporal Knowledge Graphs"
_EMNLP/2023/Conference — EMNLP 2023 Main_

### Official Review · Reviewer_juCQ · 2023-07-30

**Typos Grammar Style And Presentation Improvements:** Missing period in line 658
**Soundness:** 4

**Excitement:**

4: Strong: This paper deepens the understanding of some phenomenon or lowers the barriers to an existing research direction.

**Paper Topic And Main Contributions:**

This paper works on temporal knowledge graph completion, i.e., predicting missing links or time periods based on existing factual knowledge.  The main novelty of the proposed method is that it uses an intuitive rule language to integrate the Allen algebra relations and KG relations, which aims at enforcing temporal consistency between neighboring predicates in the rule body. The advantages of the method is that it can perform link and time interval predictions in the neuro-symbolic setting, and the prediction is more human-interpretable. Besides, the method also works well with limited data. Empirical evaluations verify these advantages.

**Questions For The Authors:**

In the end, the authors mention that they will explore combinations of NeuSTIP with box embeddings and GNN. I wonder how these can benefit each others.

**Reasons To Accept:**

1. This paper proposes the first neural-symbolic temporal KG completion (TKGC) approach named NeuSTIP that performs  both link and time-interval prediction. Existing neural-symbolic methods either focus on time-instant KGs or only do link prediction.
2.  The proposed method first considers enforcing temporal consistency between neighboring predicates in the rule body, by integrating the complete set of Allen algebra relations and KG relations. In my opinion, such modeling is novel in TKGC.
3. The authors also propose a hybrid of NeuSTIP and a KG-embedding model TimePlex (Jain et al., 2020). Such hybrid model obtains best results on both prediction tasks.
4. In general, neural-symbolic TKGE seems to be a good choice and has many advantages, e.g., it is human-interpretable, and works well with limited data.

**Reasons To Reject:**

The paper lacks a figure illustrating the overall idea of the proposed method. I can only get the overall idea after I finish the reading of the whole methdology section. It would be significantly improve the readability and presentation of the paper if the authors could add one "bigmap" figure in the paper.


**Reproducibility:**

4: Could mostly reproduce the results, but there may be some variation because of sample variance or minor variations in their interpretation of the protocol or method.

**Reviewer Confidence:**

3: Pretty sure, but there's a chance I missed something. Although I have a good feel for this area in general, I did not carefully check the paper's details, e.g., the math, experimental design, or novelty.

---

> ### Author Rebuttal · Authors · 2023-08-29
>
> Thank you for your valuable feedback. We answer your comments below:
>
> **Q1: The paper lacks a figure illustrating the overall idea of the proposed method. I can only get the overall idea after I finish the reading of the whole methodology section. It would be significantly improve the readability and presentation of the paper if the authors could add one "bigmap" figure in the paper.**
>
> Your suggestion about “bigmap” figure is excellent because it would improve the readability of the paper. We will add it to the paper. As per conference rules, we cannot provide a figure in the rebuttal but provide a high level idea of how the figure would look like.
>
> Our figure will consist of three vertical blocks. In the first block (block 1) NeuSTIP would learn the temporal logical rules from a given TKG, in the second block (block 2) it will ground the rules for a given query and also compute the rule score there. In the third block (block 3), it would compute the path score for both link and time-interval prediction and compute the final score in Equation 2.
>
> Specifically, in the first block the model would take the TKGC as input and produce the rules in Equation (1). These rules would be fed to the second vertical block in the figure. This block would also take the query as input and would ground the rules by taking rules, query and TKGC as input. From the groundings, one link (arrow) would go to block three where the path score for each path would be computed in two sub paths, one for the link prediction and one for the time prediction. The other link (arrow) from the groundings would stay in block 2, where rule score would be computed by two units GRU and PCA score. At the bottom of block 3, the final rule score and the path score will be fed to a unit that represents the score in Equation (2) and this will be set as an output of Block 3.
>
> **Q2: In the end, the authors mention that they will explore combinations of NeuSTIP with box embeddings and GNN. I wonder how these can benefit each other.**
>
> From this we mean that, in addition to ensembling NeuSTIP with Timeplex which is an embedding based model, we could also ensemble our model with other type of models such as Temporal Box Embedding model such as BoxTE[1], although this model is time-instance based model and we would need to modify the model for time interval prediction. The major benefit of this model would be that while Timeplex considers entities and relations as points in multi-dimensional hyper-space, box embeddings based models consider entities or relations as boxes in hyper-space and can hence encapsulate more information inside them. It would be interesting to see how this affects the performance of the model.
>
> We can also ensemble our NeuSTIP model with GNN based models  such as xERTE model [2]. Our proposed model benefits from Timeplex the most when a given quadruple is not covered by any rule in the proposed model. In those cases, Timeplex helps in performance improvement but it is still not using the path information because it's fundamentally an embedding based model. We could benefit from graph based models in such cases with the additional advantages being that we would  exploit paths to compute scores in such cases.
>
> References:
>
> 1.) Johannes Messner, Ralph Abboud,Ismail Ilkan Ceylan, “Temporal Knowledge Graph Completion using Box Embeddings”, AAAI 2022.
>
> 2.) Zhen Han, Pen Chen, Yunpu Ma, Volker Tresp, “Explainable Subgraph Reasoning for Forecasting on Temporal Knowledge Graphs”, ICLR 2021

---

### Official Review · Reviewer_N5Ch · 2023-08-01

**Typos Grammar Style And Presentation Improvements:** 1.Can the explanation of the model's …
**Soundness:** 4

**Excitement:**

3: Ambivalent: It has merits (e.g., it reports state-of-the-art results, the idea is nice), but there are key weaknesses (e.g., it describes incremental work), and it can significantly benefit from another round of revision. However, I won't object to accepting it if my co-reviewers champion it.

**Paper Topic And Main Contributions:**

This paper discusses work on exploring methods for temporal KGC (temporal knowledge graph completion). The paper draws the motivation for the work that most knowledge graph completion work make use of Graph algorithm like GNN, language model based solutions and knowledge-graph embedding based solutions. These methods, including the neural-symbolic approaches,  posses characteristics of unpredictable time intervals and poor human interpretability. The author/s then proposes a method that  use an intuitive rule language that integrates the complete set of allen algebra relation and KG relations, with the ability of computing confidence of temporal rules that combines both symbolic and embedding information , the author proposes a brand new NS-TKGC model which can perform time interval prediction. It also provide novel scoring function which combine rule score, path score and time information, it’s not bad. Last, the 2 benchmarks they provided, WIKIDATA12k and YAGO11k are challenging, which is good.

**Reasons To Accept:**

The idea presented in this article can be considered quite innovative. It introduces a neural-symbolic TKGC method that enhances interpretability, building upon the traditional KG embedding-based KGC approach. In comparison to other methods that utilize learning temporal logic rules and attention mechanisms, the proposed NeuSTIP model can handle two types of problems: link prediction and time interval prediction. .Also this paper discusses a novel approach for ranking candidate answers, proposing two scoring functions: one based on rule embedding and relevant rule head embedding, where the rule score is computed using the inner product. Additionally, the path score is generated using a Gaussian distribution. The combination of these scoring functions is utilized to filter the results. Furthermore, to incorporate temporal features, the final scoring function incorporates information from models such as Tcomplex embeddings. These innovative aspects represent relative significant advancements in the field. It’s might be a good paper with some novel ideas.

**Reasons To Reject:**

I am not sure if there is any “previous ideas” which share same idea with this paper.So, I just staring from the expression in this article to conduct and analysis of it’s grammar and readability.Formula 2 is quite challenging, especially when initially reading it, as the derivation details for rule embedding and path embedding are not yet understood. Consequently, the comprehension of the formula is not clear.

Considering this, it might be beneficial to rearrange the order of formulas 3, 4, and 5." Btw, is this expression method in Part 6 acceptable? I am not entirely certain about its writing style. At the end of part 6, the article mentions the possibility of mixing multiple allen relations, which may confuse the model and result in performance issues. Some Allen relations have a more significant impact on the results compared to others, but there is limited mention of experiments and validations in this regard, should further experimental evidence be supplemented.

There is another question: The best results are obtained by the combined model of NeuSTIP and Tcomplex-like embedding models. How can we determine if the experimental performance is mainly contributed by NeuSTIP? Alternatively, when combining the two models, how do they influence each other when presented with a specific question?

**Reproducibility:**

4: Could mostly reproduce the results, but there may be some variation because of sample variance or minor variations in their interpretation of the protocol or method.

**Reviewer Confidence:**

3: Pretty sure, but there's a chance I missed something. Although I have a good feel for this area in general, I did not carefully check the paper's details, e.g., the math, experimental design, or novelty.

---

> ### Author Rebuttal · Authors · 2023-08-29
>
> Thank you for finding our approach innovative, and experiments sound. We address your comments below
>
> **Q1:  I am not sure if there are any “previous ideas” which share the same idea with this paper.**
>
>
> To the best of our knowledge, our work is the second work to use Allen algebra after TILP model [2] and the first one to use all the 13 Allen predicates and show their value in the context of time-interval prediction. The formula in Equation (2) is inspired from the score function in RNNLogic[1]. However, we extend the formula to temporal setting by performing both link and time interval prediction in temporal domains.
>
> **Q2: I just staring from the expression in this article to conduct an analysis of its grammar and readability.Formula 2 is quite challenging, especially when initially reading it, as the derivation details for rule embedding and path embedding are not yet understood. Consequently, the comprehension of the formula is not clear. Considering this, it might be beneficial to rearrange the order of formulas 3, 4, and 5.**
>
> We will add the following sentences to explain the definition of rule score $(\psi(L_{j})[c])$ and path score $(\phi(path)[c])$ after the Equation (2) in the final draft of the paper: Equation (2) indicates that the candidate answer ‘c’ can be arrived at by firing multiple rules in the rule set $\mathcal{L}$ and for one rule $L_{i} \in \mathcal{L}$, multiple paths can be followed in the graph $\mathcal{G_{AW}}$ to arrive at candidate $c$. Further the importance of each rule in the final score is captured by rule score $(\psi(L_{j})[c])$ and the significance of each path in a given rule is encoded by the path score $(\phi(path)[c])$ parameter.
>
> As you know, there are two styles of describing a complex system: the top-down approach and bottom-up approaches. Generally, first giving the big picture and then going into the details is better for readability. That's why we followed the top-down approach to the writing and ordered the current equations as (2),(3),(4),(5). We will try to improve the readability by further giving further information about $\psi(L_{j})[c]$ and $\phi(path)[c]$ by giving the forward references in the paragraph that follows Equation (2).
>
> **Q3: Btw, is this expression method in Part 6 acceptable? I am not entirely certain about its writing style.**
>
> If you are pointing out to Q1 to Q6 in line 479 to 488 that we wrote the questions one after the other, then we did that to save space. If the paper gets accepted, we would use the extra space to write each question in a new line.
>
> **Q4: At the end of part 6, the article mentions the possibility of mixing multiple Allen relations, which may confuse the model and result in performance issues. Some Allen relations have a more significant impact on the results compared to others, but there is limited mention of experiments and validations in this regard, should further experimental evidence be supplemented.**
>
>  In order to answer this question of the reviewer, we performed an additional experiment.
>
> **Experiment:** It is very difficult to study the impact of Allen relations present in a given rule in our current rule language, because multiple Allen relations are present in one rule. Further, Allen relations are interleaved between the KG relations to bind the time of two neighboring KG relations in the rule body. However, the first Allen relation $\textit{a1}$ has its unique contribution to the score computation, especially in the case of Time interval prediction as discussed in the paper. Hence we  first divide Allen predicates into two categories: $\textit{DET}$ refers to the the Allen predicates which fix either the start or the end time of the time interval for e.g.: inverse_finishes, meets, inverse_meets, starts, inverse_starts, equals and $\textit{NONDET}$ refers to the Allen predicates where both ends are free, for example: before, after, overlaps, during, inverse_during, inverse_overlaps. Then we categorize the rules into two categories based upon whether the first Allen relation of the rule falls into $\textit{DET}$ or $\textit{NONDET}$ category while ignoring the rules in the other category and predicting scores in Equation (2) based on rules in a given category. The results for the link prediction and time interval prediction in two categories along will the results obtained the all rules considered ($\textit{OVERALL}$) in Table 1 and 2 of the paper are presented below
>
>
> | Dataset     | Allen Relation|  MRR     | aeIOU    |
> |-------------|---------------|----------|----------|
> |             |     DET       |  18.62   |  22.55   |
> |  Yago11k    |    NONDET     |  17.50   |  16.62   |
> |             |    OVERALL    |  23.81   |  22.49   |
> |-------------|---------------|----------|----------|
> |             |     DET       |  23.45   |  27.31   |
> | WIKIDATA12k |    NONDET     |  25.30   |  21.97   |
> |             |    OVERALL    |  31.98   |  27.36   |
> |-------------|---------------|----------|----------|
>
> Results in $\textit{DET}$ category are better than $\textit{NONDET}$ for link prediction for YAGO11k and the results of $\textit{NONDET}$ category are better than $\textit{DET}$ category for link prediction for WIKIDATA12k dataset.  For time interval prediction, $\textit{DET}$ alone can help us in achieving similar to our $\textit{OVERALL}$ model performance for time prediction. Further, for link prediction, using only one of them reduces the performance as compared to the $\textit{OVERALL}$ mode. This is because the head Allen relation does not as directly influence the entity reached by the rule during link prediction as compared to the time prediction. Impact on link prediction seems to be governed by the total number of rules, as more rules help in grounding the gold entity because paths are available to ground them. Because of this reason when the rules are divided into $\textit{DET}$ and $\textit{NONDET}$ categories, lesser numbers of rules are available for link prediction which results in drop of performance. In summary, using both sets of relations finds competitive performance in all settings.
>
> **Q5: There is another question: The best results are obtained by the combined model of NeuSTIP and Tcomplex-like embedding models. How can we determine if the experimental performance is mainly contributed by NeuSTIP? Alternatively, when combining the two models, how do they influence each other when presented with a specific question?**
>
> We can see that our NeuSTIP base model by itself gives competitive performance to the best embedding based model (TimePlex) as demonstrated in Tables (1) and (2) in the paper. If we perform ablation on any one model from the ensembled model, the performance gets worse.
>
> Furthermore, in order to answer the above question, we considered the ranks of NeuSTIP, Timeplex and an ensemble (NeuSTIP w/ KGE) individually for each test quadruple obtained during link prediction and performed a comparative study to understand how the KGE models contribute to the performance of NeuSTIP.
>
> Our major conclusion after analysis is that NeuSTIP mainly fails for the cases when no rule gets fired for a given quadruple for the gold entity resulting in zero score in Equation (2) in paper. For instance, we found
> 796 such quadruples (out of total 4102 test instances) in YAGO11k and 609 quadruples (out of 8124 quadruples in test data) in WIKIDATA12k, where our model generated zero score. For such instances, the average rank of our model was 5327 for YAGO11k, and  6307 for  WIKIDATA12k while the average rank of Timeplex for such cases was 1654.51 for YAGO11k and 1420.07 for WIKIDATA12k. Therefore our model relies on ensembling for the cases when it obtains no groundings for any rules learnt by NeuSTIP for a given quadruple.
>
> However, for the cases when the model obtains the groundings, it performs better than the corresponding Timeplex and Ensemble (NeuSTIP w/ KGE) model. For instance, when NeuSTIP obtained its grounding, its average rank is 51.1, while the average rank of Timeplex for such cases is 357.54, and is 169.70 for ensemble for YAGO11k datasets. Similarly, the average rank of NeuSTIP for cases when groundings were found is 38.09 while the average rank of Timeplex for such cases is 103.27, and ensemble for such cases is 88.65 for WIKIDATA12k. Hence we conclude that NeuSTIP relies on ensembling when it finds no grounding but outperforms the ensembling and the Timeplex model when it obtains the groundings for the candidate entity in link prediction for the rules learnt by NeuSTIP.
>
>
> **Q6: Can the explanation of the model's human interpretability be made clearer?**
>
> With human interpretability, we mean that (i) rules are human readable, (ii) we always know how (or why) we arrive at a certain answer. For example, consider the grounding of the rule provided in the Section 6:
>
> isAffiliatedTo(Franz_Dahlem, Communist_Party_of_Germany, [1920,1946]) ←During([1920,1946],[1899,1974])∧ isMarriedTo(Franz_Dahlem, Kathe_Dahlem
> ,[1899,1974])∧Contains([1899,1974],[1920,1946])∧isAffiliatedTo(Kathe_Dahlem, Communist_Party_of_Germany, [1920,1946])
>
> 1. In order to understand this grounding, we request the reviewer to first consider the head (isAffiliatedTo) and the first two relations (during and isMarriedTo) in the rule body. By considering this partial rule grounding, we can conclude that person/entity Franz_Dahlem is affiliated to party/entity Communist_Party_of_Germany during the time when he is married to person/entity Kathe_Dahlem. Here the affiliation time interval ([1920,1946]) is subsumed by the married time interval which means the entire time he was affiliated to the party, he was married.
> 2. Now consider the last two relations in the body - contains and isAffiliatedTo. Also note that the Allen relations ‘contains’ and ‘during’ are inverse of each other. This partial rule grounding implies that the entire time ‘Kathe_Dahlem’ was affiliated to Communist_Party_of_Germany, she was in her marriage time interval (the marriage time is being provided by the first argument of contains).
>
> Combining these reasonings we can conclude that Franz_Dahlem is affiliated to Communist_Party_of_Germany (the rule head) at a certain time because he was married to a woman (point 1) who was also affiliated to the same party (point 2) at around the same time interval.
>
> References:
>
> [1]  Meng Qu, Junkun Chen, Louis-Pascal A. C. Xhonneux, Yoshua Bengio, and Jian Tang. RNNLogic:  Learning Logic Rules for Reasoning on Knowledge Graphs. In ICLR, 2021.
>
> [2] Siheng Xiong, Yuan Yang, Faramarz Fekri, and James Clayton Kerce. TILP: Differentiable Learning of Temporal Logical Rules on Knowledge Graphs. In ICLR. 2023.

---

### Official Review · Reviewer_Yomj · 2023-08-05

**Soundness:** 3

**Excitement:**

3: Ambivalent: It has merits (e.g., it reports state-of-the-art results, the idea is nice), but there are key weaknesses (e.g., it describes incremental work), and it can significantly benefit from another round of revision. However, I won't object to accepting it if my co-reviewers champion it.

**Missing References:**

[1] Han, Zhen, et al. "Learning neural ordinary equations for forecasting future links on temporal knowledge graphs." Proceedings of the 2021 conference on empirical methods in natural language processing. 2021.

**Paper Topic And Main Contributions:**

Link prediction and time prediction are among the popular solutions for temporal knowledge graph completion. Previous works are categorized as embedding-based, multi-hop reasoning-based, and rule-based approaches where most of them perform link prediction and a few of them address the problem of time prediction, especially prediction of time intervals.
This paper proposes a rule-based approach to perform both link prediction and time-interval prediction on temporal knowledge graphs. It first introduces Allen algebra as a formal system representing relations between time intervals.  To mine the rules, it first finds the ground paths from subject to object and then converts them to first-order rules.
The score of each candidate's answer (entity, start time, end time) is obtained based on two individual scores: rule score, and path score. The rule score itself is based on embedding with the gated recurrent unit and also statistical measure-based scores. Path score is additionally used for time interval prediction by modeling time gaps between head and body relations using Gaussian distribution.
Experimental results on Wikidata12k and Yago11k show the proposed method outperforms other competitors in link and time interval prediction, especially by ensembling with KGE models.

**Questions For The Authors:**

Question A: How can the model predict future links?
Question B: what is the computational complexity of different parts of the model?

**Reasons To Accept:**

--The model is founded on top of Allen algebra which is a well-defined system for modeling temporal relations.

-- The proposed approach can utilize temporal rules and combine them with embedding to perform link prediction.

-- The proposed method is designed by a chain of reasonable steps.

**Reasons To Reject:**

-- The performance of the core of the method (NeuSTIP-base) is mainly lower than the corresponding competitors. The most performance gain is obtained by the ensembling approach. In this case, it is essential to know how will be results of other models after being ensembled with TimePlex. Besides,  as the ensemble version of the model is the best model, it is important to compare it with other ensemble approaches.

-- It is not clear if the obtained rules are statistically rich and also how is it computationally expensive to use all walks.

-- The model may be influenced by low-quality paths when the path score is set to 1 for link prediction. Not all paths are informative and helpful for prediction.

-- It is not clear why only r_h and r_1 are considered in equation 5, and why other relations in the path are not considered.

-- A time distribution given each relation is modeled by a Gaussian. However, a relation may present different frequencies in different time periods and the model may not capture various degrees of temporality with a single Gaussian distribution.

-- The writing of some parts of the paper should be improved, e.g., Page 5, like 402, what is vol, or section 4.2.2 should be improved to be clearer.

-- There are several temporal KGEs e.g., [1] that can handle inductive settings in both time (future link prediction) and entity. Those methods might be compared to the proposed method in table 5.

**Reproducibility:**

4: Could mostly reproduce the results, but there may be some variation because of sample variance or minor variations in their interpretation of the protocol or method.

**Reviewer Confidence:**

3: Pretty sure, but there's a chance I missed something. Although I have a good feel for this area in general, I did not carefully check the paper's details, e.g., the math, experimental design, or novelty.

---

> ### Author Rebuttal · Authors · 2023-08-29
>
> Thank you for your review and feedback on our paper. We address your comments below.
>
> **Q1:  The performance of the core of the method (NeuSTIP-base) is mainly lower than the corresponding competitors. The most performance gain is obtained by the ensembling approach. In this case, it is essential to know how will be results of other models after being ensembled with TimePlex. Besides, as the ensemble version of the model is the best model, it is important to compare it with other ensemble approaches.**
>
> **Q1, Part 1: The performance of the core method (NeuSTIP-base) is mainly lower than the corresponding competitors.**
>
> **Link prediction:** The best baseline models (no additional gadgets) that can be directly compared against our model in Table 1 are TILP-base in the Neuro-Symbolic Temporal models category and Timeplex (base) in the embedding-based models category. When compared to TILP-base, we observe that NeuSTIP (base) model is competitive on WIKIDATA12k dataset and is 5 pt and more than 4 pt better than TILP-base on MRR and Hits@10 on YAGO11k dataset.  When compared with Timeplex (Base), our base model is still performing better than Yago11k on all metrics, however, we agree with the reviewer that it performs slightly worse than Timeplex (base) on WIKIDATA12k dataset. We wish to point out that it has been noticed in the past literature that when directly compared, embedding based models outperform Neuro-Symbolic models because neuro-symbolic models sometimes suffer from low-coverage of rules which results in zero score for some triples in KG [3].
>
> **Time prediction:** Our base model outperforms Timeplex (base), the best baseline model, on both the datasets for time-interval prediction. In fact, for YAGO11k dataset, aeIOU score of our model is 8 point higher than Timeplex (base)’s score.
>
> **Part 2: The most performance gain is obtained by the ensembling approach. In this case, it is essential to know how will be results of other models after being ensembled with TimePlex. Besides, as the ensemble version of the model is the best model, it is important to compare it with other ensemble approaches.**
>
> The reviewer has raised a valid point. We experimented (results in table below) with the state-of-the-art baseline model TILP by ensembling it with the embedding based approach Timeplex (the same we used for our ensembled approach) for different values of eta hyperparameter.  We observed a different trend compared to what we noted in NeuSTIP, i.e., the ensembling did not improve the performance of TILP (reported in table below). This counterintuitive observation points towards the sophisticated process  of ensembling.
>
> | ETA     |  10E-3   |  10E-5  |  10E-6   |  10E-8  |   0     |
> |---------|----------|---------|----------|---------|---------|
> |  H@1    |   10.09  |  11.16  |   11.26  |   11.80 |  16.81  |
> |  H@10   |    14.19 |  16.86  |   17.47  |   18.79 | 41.39   |
> |  MRR    |  11.69   |  13.24  |   13.50  |   14.25 | 25.32   |
>
>
> **Other Ensembling Approaches:** Right now we have ensembled NeuSTIP (base) with Timeplex linearly wherein score_ensemble = score(NeuSTIP) + eta*score(Timeplex) where eta is a learnable parameter. We could also explore more advanced ensemble approaches such as in Rivas-Barragan [4] that exploit the distribution of the score to ensemble the models together. We could also learn eta itself by exploiting the distribution of the score as in [5]. We would perform some experiments on different ensembles in the final draft of the paper.
>
> **Q2: It is not clear if the obtained rules are statistically rich and also how is it computationally expensive to use all walks.**
>
> **Statistical Richness of the rules:** We compute multiple metrics such as number of rules, average number of groundings per rule as a measure for statistical richness of the rules. The proposed methodology of performing all walks in NeuSTIP generated 8186 rules for YAGO11k and  31,807 rules for WIKIDATA12k dataset. The reason for obtaining a moderate number of rules while we consider all walks on TKGC is the sparsity of temporal KGCs as already discussed in the paper (section 4.1). Further, the average number of groundings per rule considering the train set is 17.87 for YAGO11k, and 885.21 for WIKIDATA12k, which is sufficiently high.
>
> **Computationally Expensive:** Total Run-time required by our model during the rule generation step for YAGO11k is 16.49s and for WIKIDATA12k is 278.68s on a single CPU of Linux server. The configuration of the Linux server where the rules were generated is Intel(R) Xeon(R) Gold 6142 CPU @ 2.60GHz.
>
> **Q3: The model may be influenced by low-quality paths when the path score is set to 1 for link prediction. Not all paths are informative and helpful for prediction.**
>
> We agree with the reviewer that assigning path score as 1 to all the path during link prediction does not distinguish the informative paths from the futile paths. In fact, motivated by RNNLogic [2], we considered assigning a real number score between 0 and 1 to each path. RNNLogic extends the RotatE model to paths by composing the embeddings of all the relations along the path. However,  finding a suitable ‘compositional’ embedding model in the presence of time-intervals is a research task in itself, as past literature on this is limited.
>
> Also, please note that for a fixed rule, the entity reached by a higher number of paths would be ascribed a higher confidence. The cumulative score also includes the rule score (Equation 2).
>
> **Q4: It is not clear why only $r_{h}$ and $r_{1}$ are considered in equation 5, and why other relations in the path are not considered.**
>
> The primary reason for learning the normal distribution between $r_{h}$ and only $r_{1}$ can be attributed to the rule language designed in Equation (1). As can be seen from the language, the target relation $r_{h}$ shares its subject only with the $r_{1}$ relation and hence considering $(r_{h}, r_{1})$ leads to learning the normal distribution between two relations that share a subject entity. For example let one grounding of head $r_{h}$ be $\textit{(Joe Biden, Born in, USA, 1942)}$ and further $r_{1}$ has grounding $\textit{(Joe Biden, Graduated From, University of Delaware, 1965)}$, then Gaussian distribution would consider the difference (1942-1965 = -23) as one of the point to compute the mean $\mu_{r_{h},r_{1}}$ in the normal difference only because both the relations contain the information about a shared entity - Joe Biden.
>
> **Q5:  A time distribution given each relation is modeled by a Gaussian. However, a relation may present different frequencies in different time periods and the model may not capture various degrees of temporality with a single Gaussian distribution.**
>
> Yes, we agree with the reviewer that a single Gaussian distribution may not capture the different frequencies in different time periods. We can, in practice, use mixture models to capture the various degrees of temporality, but that would add to the parameters and the model complexity. Exploring more ways to assign the scores to the candidate start and end time instances obtained after Allen predicate filtering is an avenue for further research.
>
> **Q6: The writing of some parts of the paper should be improved, e.g., Page 5, like 402, what is vol, or section 4.2.2 should be improved to be clearer.**
>
>  We thank the reviewer for pointing out the typos in the paper. We will fix them in the final version of the paper. Vol in line 402 refers to the size of the time interval. $(T^{ev} ⋂ T^{pr})$ refers to overlap in time-interval [2]. $T^{ev}⋓T^{pr}$ is the smallest single contiguous interval (hull) containing all of $T^{ev}$ and $T^{pr}$ .E.g.,[1,2] ⋓ [30,40]=[1,40].
>
> We request the reviewer to help us understand the part of section 4.2.2 where the writing needs to be improved. It is quite clear to us. We will make the best effort to improve the understandability of section 4.2.2 in the final draft of the paper, if we understand the confusion. For better understanding of the reviewer, we explain the high-level idea of Section here: In path score computation for time interval prediction, we score the start time $t_{b}$ and end time $t_{e}$ independently. To compute the path score of start time, we first compute the mean and the variance of the difference of start time between $r_{h}$ and $r_{1}$ for the cases where they share the subject entity. Then we compute the normal distribution between difference of time $t_{2b}$ which is obtained by partial grounding of a given rule and one of the potential answer $t_{1b}$ for the start time and return this normal distribution as the score $\phi(path)[t_{1b}]$.
>
> **Q7: There are several temporal KGEs e.g., [1] that can handle inductive settings in both time (future link prediction) and entity. Those methods might be compared to the proposed method in table 5.**
>
> In our Table 5, we perform inductive link prediction, but without considering forecasting tasks. Model [1] seems to only work for the link forecasting case. So, it can’t be directly applied to our experimental setting of Table 5 – hence our model can not be directly compared to the TANGO model [1].
>
> **Question A: How can the model predict future links?**
>
> On your suggestion, we performed link forecasting on NeuSTIP(base) whose experimental setup and the results are explained below:
>
> **Experimental Setup:**  We sorted YAGO11k/WIKIDATA12k datasets according to the start time of the time intervals, to make it suitable for forecasting ensuring the time in train and the test sets are disjoint as explained soon.
>
> **Running NeuSTIP (base):**
>
> 1.) For the rule generation phase, we ensure that the start time of any interval occurring in the body of the rule is strictly less than that of the quadruple used in the head. Consequently, our model finds fewer rules.
>
> 2.) Similarly during grounding the rules, while performing parameter learning, we maintain the above condition.
>
> 3.) While running inference on the test set, this condition is automatically ensured due to the way in which the dataset is split. It is important to note that due to fewer rules and lesser groundings found in this split, our model's performance also reduces.
>
> **The time range of datasets for link forecasting:**
>
> We list the possible values of the start time for train, validation and test phases in YAGO11k/WIKIDATA12k  datasets below:
>
> YAGO11k:
>
> Train start times range : [T_min, 2007]
>
> Valid start times range : [2008, 2011]
>
> Test start times range: [2012, T_max]
>
> WIKIDATA12k:
>
> Train start times range : [T_min, 2009]
>
> Valid start times range : [2010, 2012]
>
> Test start times range: [2013, T_max]
>
> **Dataset size:**
>
> The size of dataset (number of quadruples) for YAGO11k after this split comes out to be (without adding inverses):  Train: 16,806, Valid: 1823, Test: 1880
>
> The size of dataset (number of quadruples) for WIKIDATA12k after this split comes out to be (without adding inverses): Train: 33,368, Valid: 3649, Test: 3604
>
> **Results of NeuSTIP (base) for link forecasting:**
>
> We obtained the final results of link forecasting by employing time aware filtering [2] that we have employed in all the experiments in our paper. The results are given below:
>
> YAGO11k:         MRR: 7.08, H@1: 3.38, H@10: 13.86
>
> WIKIDATA12K: MRR: 13.82, H@1: 7.45, H@10: 26.07
>
> **Link Forecasting for TANGO model[1]:**
>
> We also obtained results of link forecasting for the TANGO model[1] on the YAGO11k/WIKIDATA12k datasets. Please note that the results obtained below are not directly comparable to the results of link forecasting to NeuSTIP results provided above because NeuSTIP is a time-interval dataset and TANGO is a time-instance dataset, their time-aware filtering techniques are different from each other. We are providing these results only for completeness.
>
> **Experimental Setup:**
>
> 1.) Here, we consider the start time of all the quadruples, since the TANGO model is inherently designed for time instance datasets and cannot directly work with Time Interval datasets as our model.
>
> 2.) We pick the code from https://github.com/TemporalKGTeam/TANGO, and run it on the default set of hyperparameters after pre-processing the datasets in exactly the same way as which the model does for ICEWS05-15 dataset.
>
> **Results for Link forecasting for TANGO Model:**
>
> Below, we provide the results of applying the TANGO model for link forecasting on YAGO11k/WIKIDATA12k datasets for different filtering settings.
>
>  YAGO11k (Raw):                               MRR: 2.073,  H@1: 0.638,  H@10: 4.681
>
>  YAGO11k (Time Aware filter):           MRR: 2.089, H@1: 0.665, H@10: 4.681
>
>  YAGO11k (Time Unaware filter):       MRR: 2.726, H@1: 1.197, H@10: 5.346
>
> WIKIDATA12k (Raw):                         MRR: 4.718, H@1: 2.79, H@10: 8.075
>
> WIKIDATA12k (Time Aware filter):     MRR: 4.83, H@1: 2.917, H@10: 8.131
>
> WIKIDATA12k (Time Unaware filter): MRR: 6.39, H@1: 4.735, H@10: 8.963
>
> For the definition of the RAW, time aware and Time unaware filtering please refer to TANGO model [1].
>
> **NeuSTIP vs TANGO comparison:** Lastly, we obtained the results of NeuSTIP for link forecasting(LF) for raw (without filtering) settings so that the results of TANGO and NeuSTIP can be compared directly for raw settings.
>
> NeuSTIP link forecasting results for WIKIDATA12k (Raw): MRR: 9.455, H@1: 0.0, H@10: 23.557
>
> TANGO link forecasting results for WIKIDATA12k (Raw):   MRR: 4.718, H@1: 2.79, H@10: 8.075
>
> NeuSTIP link forecasting results for YAGO11k (Raw):        MRR: 5.289, H@1: 0.0, H@10: 12.978
>
> TANGO link forecasting results for YAGO11k (Raw):      MRR: 2.073,  H@1: 0.638,  H@10: 4.681
>
> As can be observed, our model outperforms TANGO for link forecasting on both YAGO11k and WIKIDATA12k on MRR and H@10.
>
> **Question B: What is the computational complexity of different parts of the model?**
>
> This is the complexity analysis for various components in our model:
>
>  1.) **Rule Extraction:** Here, the time taken is essentially equal to the number of walks of length less than or equal to max rule length $L$ in the Knowledge Graph. The complexity of this is $N_{walk}(1) + N_{walk}(2) + … N_{walk}(L)$. Since at every entity $\textit{E}$ occurring in the walk, we explore all its neighbors, this can be loosely bound to $G_{train}(\delta)^{L}$, where $\delta$ is the maximum degree of an entity.
>
> 2.) **Finding rule groundings for Parameter Learning:**
>
> Let $N_{rule}$ be max number of rules for a tuple in $G_{train}$, $N_{path}$ be maximum number of paths following a given rule for a quadruple.
>
> Time Complexity for grounding for Link prediction comes out as $G_{train}N_{rule}N_{path}L$
>
> For time prediction, the query is $(h,r,t,?)$. Essentially for every path, there is some corresponding starting time interval $E$ occurring in the first predicate of the rule body. Maximum such $E$ would be bounded by the number of tuples of the form $(h, r_{body1}, \*, \*)$, which is bounded by the degree of node $h$. Among the paths, we only select those ones which end at the entity $t$.
>
> For each $E$, we use the first Allen predicate in the body of the rule to perform filtering, let $T_{startmax}$ and $T_{endmax}$ be the maximum number of possible candidates for any $E$ after Allen filtering.
>
> For each such candidates, we use the distribution scores, which are computed only once for a fixed difference and a fixed pair of relations, (essentially this component would contribute $O(TR^2)$ time)
>
> Overall Time complexity is given as $G_{train}N_{rule}(N_{path}L + \delta(T_{startmax} + T_{endmax}) ) + TR^2$
>
> Time taken to pre-compute pairwise difference distribution:
>
> For a fixed entity $E$, all the facts would be considered in pairs with each other, which will contribute to some pair of $(r1,r2)$.
> Hence the total complexity is given as $O(E(\delta)^2)$
>
> 3.) **Time taken for PCA score**:
>
> Given a quadruple $(h,r,?,T)$ for link prediction, we obtain all the set of reached entities following all possible rules. Among these, the set of reached entities which are an answer of $(h,r,?,T)$ will contribute to the numerator of PCA, rest will contribute to the denominator of PCA.
> Let $N_{cand}$ be the maximum number of answer entities.
>
> We essentially maintain a mask for each query of the size of $E/T$, which is set to 1 for the answer entities/instances, and this is used in computing the PCA score.
>
> Link complexity is given as $O(G_{train}(E + N_{rules}N_{cand}))$
>
> Similarly for PCA of time, complexity comes out as:  $O(G_{train}(T + N_{rules} (T_{startmax} + T_{endmax}))$
>
> 4.) **Time Taken for Inference** :
>
> We obtain rule score from GRU, find path score through the groundings , let $N_{path}$ be the maximum number of paths for a given query in the background KB by following a rule.
>
> Link complexity is given as $G_{test}N_{rule}N_{path}L$
>
> Time complexity is given as $G_{test}N_{rule}(N_{path}L +\delta*(T_{startmax} + T_{endmax}) )$
>
> For link, time prediction, we get the scores using the groundings and aggregating on all rules (where rule scores are obtained by pre-computed PCA, passing through GRU).
> In Link Prediction, we simply produce a ranked list of entities using the obtained scores. ($(O(E))$ time per query)
>
> For Time Prediction, after obtaining the scores, we take $O(T^{2})$ time to predict the best interval per query.
>
> References:
>
> [1] Zhen Han,Zifeng Ding,Yunpu Ma,Yujia Gu, Volker Tresp, “Learning Neural Ordinary Equations for Forecasting Future Links on Temporal Knowledge Graphs”, EMNLP 2021
>
> [2] Prachi Jain, Sushant Rathi, Mausam, Soumen Chakrabarti, “Temporal Knowledge Base Completion: New Algorithms and Evaluation Protocols”, EMNLP 2020
>
> [3] Ananjan Nandi, Navdeep Kaur, Parag Singla, Mausam, “Simple Augmentations of Logical Rules for Neuro-Symbolic Knowledge Graph Completion”, ACL 2023 (Short Paper)
>
> [4] Daniel Rivas-Barragan and others, Ensembles of knowledge graph embedding models improve predictions for drug discovery, Briefings in Bioinformatics, Volume 23, Issue 6, November 2022
>
> [5] Yinquan Wang,Yao Chen, Zhe Zhang, Tian Wang, “A probabilistic ensemble approach for knowledge graph embedding”, Neurocomputing,volume 500, 2022

---

### Meta-Review · Area_Chair_2KfJ · 2023-09-22

**Recommendation:** 4

**Metareview:**

Reviewers agreed that this paper presents an innovative approach to temporal knowledge graph completion. The method's combination of Allen intervals, neurosymbolic scoring, interpretable rule mining were recognized as strong contributions. Sufficient detail was provided to be able to reproduce the results. Several reviewers pointed out issues with presentation, writing, and clarity that could be improved. While the absolute performance of the proposed method for KGC in isolation does not represent a new SotA result, time interval prediction does improve over prior models and the authors propose an ensembling approach to improve KGC quality. During rebuttal discussion, the authors provided ancillary results that helped to quantify improvements and provide details to support their conclusions. The ideas in this paper and its goal of predicting both time intervals and tail entities are likely to generate significant interest for a small community interested in TKG modeling.

---

### Decision · Program_Chairs · 2023-10-07

**Decision:**

Accept-Main

**Comment:**

Reviewers agreed that this paper presents an innovative approach to temporal knowledge graph completion. The method's combination of Allen intervals, neurosymbolic scoring, interpretable rule mining were recognized as strong contributions. Sufficient detail was provided to be able to reproduce the results. Several reviewers pointed out issues with presentation, writing, and clarity that could be improved. While the absolute performance of the proposed method for KGC in isolation does not represent a new SotA result, time interval prediction does improve over prior models and the authors propose an ensembling approach to improve KGC quality. During rebuttal discussion, the authors provided ancillary results that helped to quantify improvements and provide details to support their conclusions. The ideas in this paper and its goal of predicting both time intervals and tail entities are likely to generate significant interest for a small community interested in TKG modeling.